# Lineage mapping identifies molecular and architectural similarities between the larval and adult *Drosophila* central nervous system

Haluk Lacin*, James W Truman

Janelia Research Campus, Howard Hughes Medical Institute, Ashburn, United States

**Abstract** Neurogenesis in *Drosophila* occurs in two phases, embryonic and post-embryonic, in which the same set of neuroblasts give rise to the distinct larval and adult nervous systems, respectively. Here, we identified the embryonic neuroblast origin of the adult neuronal lineages in the ventral nervous system via lineage-specific GAL4 lines and molecular markers. Our lineage mapping revealed that neurons born late in the embryonic phase show axonal morphology and transcription factor profiles that are similar to the neurons born post-embryonically from the same neuroblast. Moreover, we identified three thorax-specific neuroblasts not previously characterized and show that HOX genes confine them to the thoracic segments. Two of these, NB2-3 and NB3-4, generate leg motor neurons. The other neuroblast is novel and appears to have arisen recently during insect evolution. Our findings provide a comprehensive view of neurogenesis and show how proliferation of individual neuroblasts is dictated by temporal and spatial cues.

*For correspondence: lacinh@ janelia.hhmi.org

**Competing interests:** The authors declare that no competing interests exist.

## Introduction

The embryonic ventral nerve cord (VNC) of *Drosophila* has been used as a model system for over three decades to understand how a small number of neuronal stem cells, called neuroblasts (NBs), generate a highly complex but organized tissue in which almost all cells adopt unique fates (*Jimenez and Campos-Ortega, 1979*; *Cabrera et al., 1987*; *Doe, CQ 1992*; *Skeath and Carroll, 1992*; *Bossing et al., 1996*; *Schmidt et al., 1997*; *1999*; *Rickert et al., 2011*). Patterning of the neural ectoderm is the first step in promoting neuronal diversity. The orthogonal interaction of segment-polarity genes [e.g., *runt (run)*, *wingless (wg)* and *gooseberry (gsb)*] and columnar genes [e.g., *ventral nervous system defective (vnd)*, *intermediate neuroblasts defective* (*ind*), and *muscle specific homeobox* (*msh*; also referred to as Drop)] divides the neuroectoderm into a Cartesian grid system, in which each NB acquires a unique identity based on its position within the grid (reviewed in *Skeath, 1999*). About 30 distinct NBs form in a segmentally repeated bilateral pattern through most of the VNC segments, although the number of NBs is reduced in the anterior gnathal and terminal abdominal segments (*Bossing et al., 1996*; *Schmidt et al., 1997*; *1999*; *Technau et al., 2014*; *Birkholz et al., 2013*). Each NB undergoes multiple rounds of asymmetric cell division. During each division it renews itself and generates a secondary precursor cell, called a ganglion mother cell (GMC), which terminally divides to generate a pair of neurons or glia (*Campos-Ortega, 1993*; *Goodman and Doe, 1993*; *Rhyu et al., 1994*; *Spana et al., 1995*). Through successive cell divisions, the number of which depends on the NB identity, each NB produces unique and highly diverse progeny (*Bossing et al., 1996*; *Schmidt et al., 1997*; *1999*). Recent studies have shown that many NBs in the embryonic VNC undergo the following temporal changes of the transcription factor

**eLife digest** Fruit flies undergo a process called metamorphosis in which they change from a maggot or larva into an adult fly. These two life stages look and behave differently and appear to have strikingly different nervous systems. The relationship between the two nervous systems has been most extensively studied in the ventral nerve cord (which is the equivalent to the spinal cord in humans). Although the ventral nerve cords of a larva and an adult fly look quite different, they are generated by the same set of stem cells known as neuroblasts. This is made possible because the neuroblasts proliferate in two separate phases: the first phase occurs in the embryo to generate the neurons of the larval nervous system, and the second phase occurs in the larva to generate neurons for the adult's nervous system.

Now, Lacin and Truman have paired each of the neurons in the adult fruit fly's nerve cord with their corresponding neurons in the nerve cords of fruit fly larvae. This involved identifying the original neuroblasts that gave rise to each of the neurons in both larval and adult fruit flies. The results suggest that most neurons that arise from a given neuroblast produce a similar set of molecules and extend similar nerve fibers, even though they work in two different nervous systems. Since neuroblasts in non-metamorphosing insects proliferate continuously, these findings also suggest that, when metamorphosis evolved, a pause was introduced to create the two separate phases of proliferation without a big effect on the types of neurons generated.

Lacin and Truman then went on to discover three neuroblasts that appear to be unique to the middle (or thoracic) segments of a fruit fly. The experiments reveal that the presence of these neuroblasts depended on specific genes that control the development of animal body plans. Two of these neuroblasts generate the so-called motor neurons that control the movement of a fly's legs. Flies only have legs on their thoracic segments, so this indicates that the development of new neurons is coordinated with the development of the body plan at the stem cell level. The third neuroblast generates neurons that connect with the leg motor neurons, and Lacin and Truman propose that this neuroblast arose from a copy of a neighboring stem cell. The resulting extra neurons may have enabled finer control over the leg movements required for activities such as walking and grooming.

Following on from this work, it is now possible to investigate how molecular events that occur from the embryonic to the adult stages of a fruit fly's life control the formation and function of its nervous system.

expression: Hunchback → Kruppel→ Pdm→ Castor (*Kambadur et al., 1998*; *Brody and Odenwald 2000*; *Isshiki et al., 2001*; *Pearson and Doe 2003*; *Grosskortenhaus et al. 2005*). Each of these factors defines a temporal identity window for the NB, and each is maintained in the GMC, establishing different transcriptional states. The GMC then divides via Notch-mediated asymmetric cell division to produce two sibling cells with distinct identities: the Notch-ON "A" cell and the Notch-OFF "B" cell (reviewed in *Jan and Jan, 2000*). Consequently, diversity within a NB lineage is produced through two main mechanisms: transcriptional changes in the NB that occur as the stem cell divides and Notch mediated asymmetric cell fates of the daughters of the GMC.

Towards the end of embryogenesis, most NBs in the thoracic and gnathal segments enter a mitotically quiescent state, whereas most NBs in the abdominal segments and a few in the thoracic segments die through apoptosis (*Peterson et al., 2002*; *Cenci and Gould, 2005*; *Baumgardt et al., 2009*). The quiescent NBs re-enter the cell cycle at the beginning of the second larval instar stage and continue to generate progeny from the Castor (Cas) window (*Tsuji et al., 2008*; *Maurange et al., 2008*). This quiescent state divides the neurogenesis of *Drosophila*, and other insects that undergo complete metamorphosis, into two phases: an embryonic phase, which generates the neurons of the larval nervous system, and a postembryonic phase, which generates adult-specific neurons. Because of the NB quiescence and the anatomical changes that occur in the CNS during late embryogenesis, it has been difficult to establish the correspondence between the embryonic and postembryonic lineages. Recently, using the technique of Flybow, *Birkholz et al. (2015)* reported the correspondence between the embryonic and postembryonic lineages. We attempted a

similar linkage using a suite of genetic and molecular tools to identify individual NB lineages in the embryo and then used these tools to bridge the postembryonic lineages to their embryonic origins. While we have concordance with most of the findings of *Birkholz et al. (2015)*, we differ on eight of the lineages. We also find that within a lineage, the postembryonically born neurons show significant similarities to neurons that are born in the embryonic Cas window in terms of axonal projection and transcription-factor expression. Moreover, our findings complete previous work on identifying embryonic and postembryonic progeny of NBs by characterizing thorax-specific NBs, NB2-3 and NB3-4, which produce leg motor neurons and identifying a novel NB, NB5-7. Our complete lineage map, and the reagents we generated to follow individual lineages throughout development, lay the groundwork for investigating how neural patterning and NB identity in the embryonic CNS direct the formation of neural circuits in the adult.

## Results and discussion

### Relationship of the embryonic and postembryonic lineages for the VNC

We linked the embryonic NBs to their postembryonic progeny via a recently developed technique, which irreversibly marks the complete progeny of a NB after the onset of GAL4 expression (*Figure 1A,B*; *Awasaki et al., 2014*). To use this technique (which we call here "reporter immortalization"), we visually screened publicly available databases (*Manning et al., 2012*; *Kvon et al., 2014*) to identify GAL4 lines whose reporter expression is restricted to one or a few embryonic NBs. We identified over 100 such GAL4 lines. Only a few of them marked an individual NB, while most marked a few NBs with or without their progeny (*Table 1*). To identify which NB lineages are marked by these GAL4 lines, we generated random lineage clones for each line (*Lacin et al., 2009*; *Nern et al., 2015*) and compared their morphology and molecular-marker expression to previously published embryonic neuronal lineages (*Bossing et al., 1996*; *Schmidt et al., 1997*; *1999*; *Birkholz et al., 2013* and references therein). With this information, we also intersected split GAL4 combinations to restrict overlapping expression patterns of different drivers to individual NBs and their progeny.

Ultimately, the GAL4 and split-GAL4 lines characterized during this study mark 28 out of 31 previously documented NBs individually (12 NBs) or in combination with a small number of other lineages (*Table 1*, examples in *Figure 2*). Many lines also drive reporter expression in the progeny of the marked NB, and their expression pattern is maintained into early larval stages.

Due to the large number of lineages and the repetitive nature of the lineage-tracing method, we will discuss only a few NBs in detail to illustrate how we identified their postembryonic progeny. We will also focus on where our results differ from *Birkholz et al. (2015)*. All of the driver lines and molecular markers that were used to link NBs to their postembryonic progeny can be found in *Tables 1* and *2*.

### NB2-4, NB2-5, NB3-5 and NB6-2 generate postembryonic lineages 18, 17, 9 and 19, respectively in the dorsal part of the nerve cord

The postembryonic lineages that are located dorsal to the neuropil include an anterior triplet, consisting of lineages 9, 17 and 18, and the posteriorly located lineage 19 (*Truman and Bate, 1988*; *Truman et al., 2004* and 2012; *Figure 3B*). In the late embryonic CNS, Dpn staining, which marks all NBs, shows the same "3+1" pattern, suggesting that these NBs generate the dorsal postembryonic lineages (*Figure 3A*). A Dichaete (D)-expressing NB is always located medially in the anterior NB triplet. We identified this NB as NB2-4 based on morphological and molecular criteria: it is marked by *eagle* (*eg*)-GAL4, *mirror*-lacz and Msh (*Figure 3C* and not shown), which are co-expressed in NB2-4 (*Higashijima et al, 1996*; *Broadus et al., 1995*; *Isshiki et al., 1997*). The morphology of lineage clones from this NB also matches the previously published features of the NB2-4 lineage: dorsolateral location, presence of a motor neuron with contralateral projection across the Anterior Commissure (AC) and into the anterior root of the Intersegmental Nerve (ISN), and the presence of interneurons that bundle with this motor neuron to cross the midline. (*Figure 3C,J*; *Schmidt et al., 1997*; *1999*). R65G02-GAL4 marks NB2-4 and occasionally a couple other NBs (*Figure 3D* and not shown). Reporter immortalization of the progeny of NB2-4 consistently labels lineage 18 (21 of 30 hemisegments) showing that NB2-4 generates lineage 18 (*Figure 3E*). In addition, D and Msh

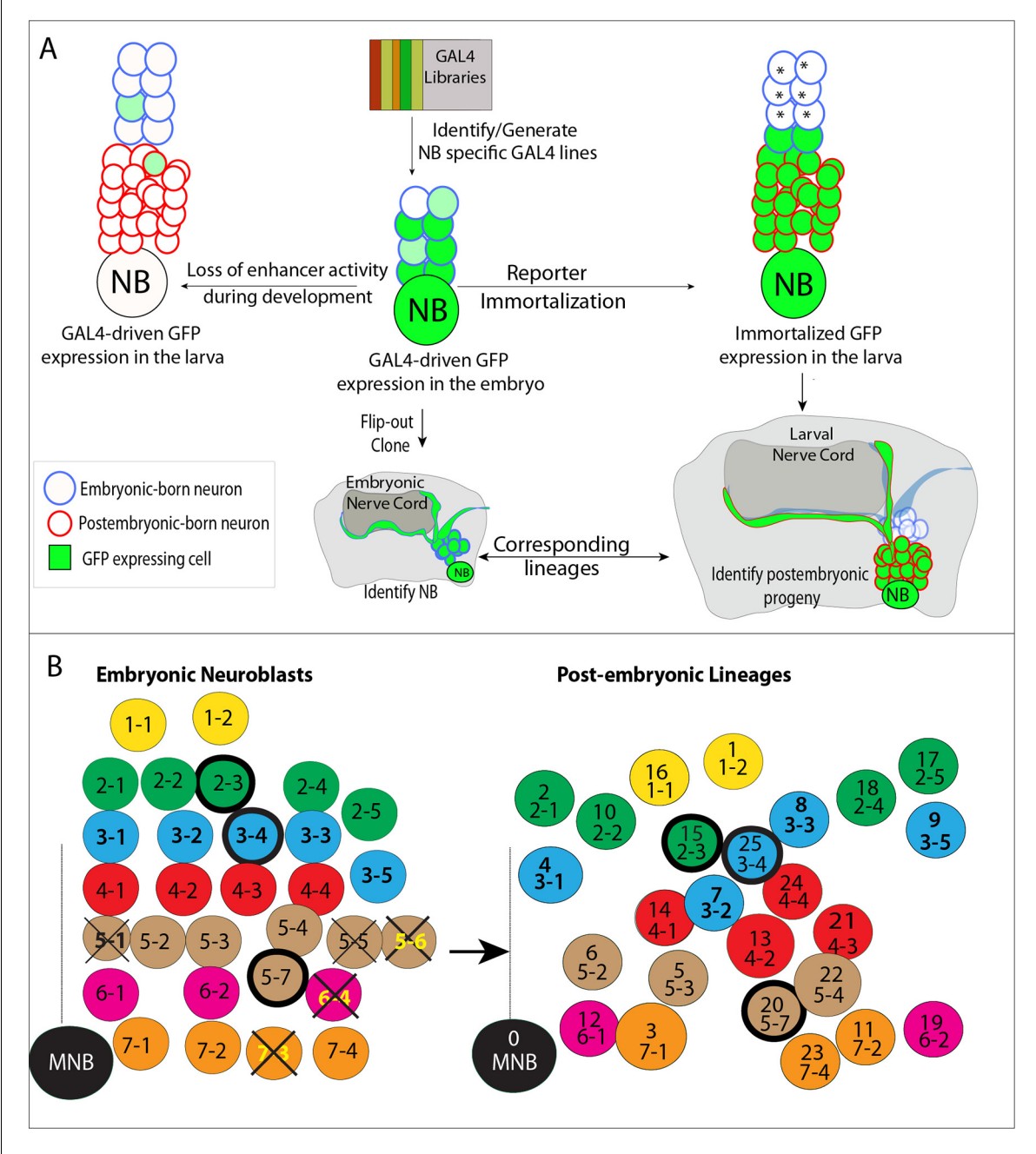

**Figure 1.** Tracing individual NB lineages identifies postembryonic progeny of NBs. (**A**) Schematic illustration of the strategies used in this study to trace NB lineages. Since expression of GAL4 lines is usually not maintained throughout the development, GAL4 expression was immortalized in progeny of the NBs to identify their postembryonic progeny. The "reporter immortalization" technique requires several steps of transcriptions and recombinations (*Awasaki et al., 2014*). Thus, cells that are born from initial divisions (marked by asterisk) after the GAL4 presence are not labeled with this technique. (**B**) Schematic representations of embryonic NBs (left) and their corresponding postembryonic lineages (right) for the T2 segment shown. 30 bilaterally symmetric NBs and 1 medial NBs generate 26 postembryonic lineages. For segment specific differences, see *Figure 1—figure supplement 1*. Row identity of NBs shown in a color code. Three thoracic specific NB lineages are outlined with a thick line. Thick crosses depict NBs, which are eliminated by apoptosis; thin crosses depict NBs, which are present at early stage embryos, but not detected at stage 17 embryos. Dashed line indicates the midline.

The following figure supplement is available for figure 1:

**Figure supplement 1.** Schematic representations of postembryonic lineages in different segments of the nerve cord shown.

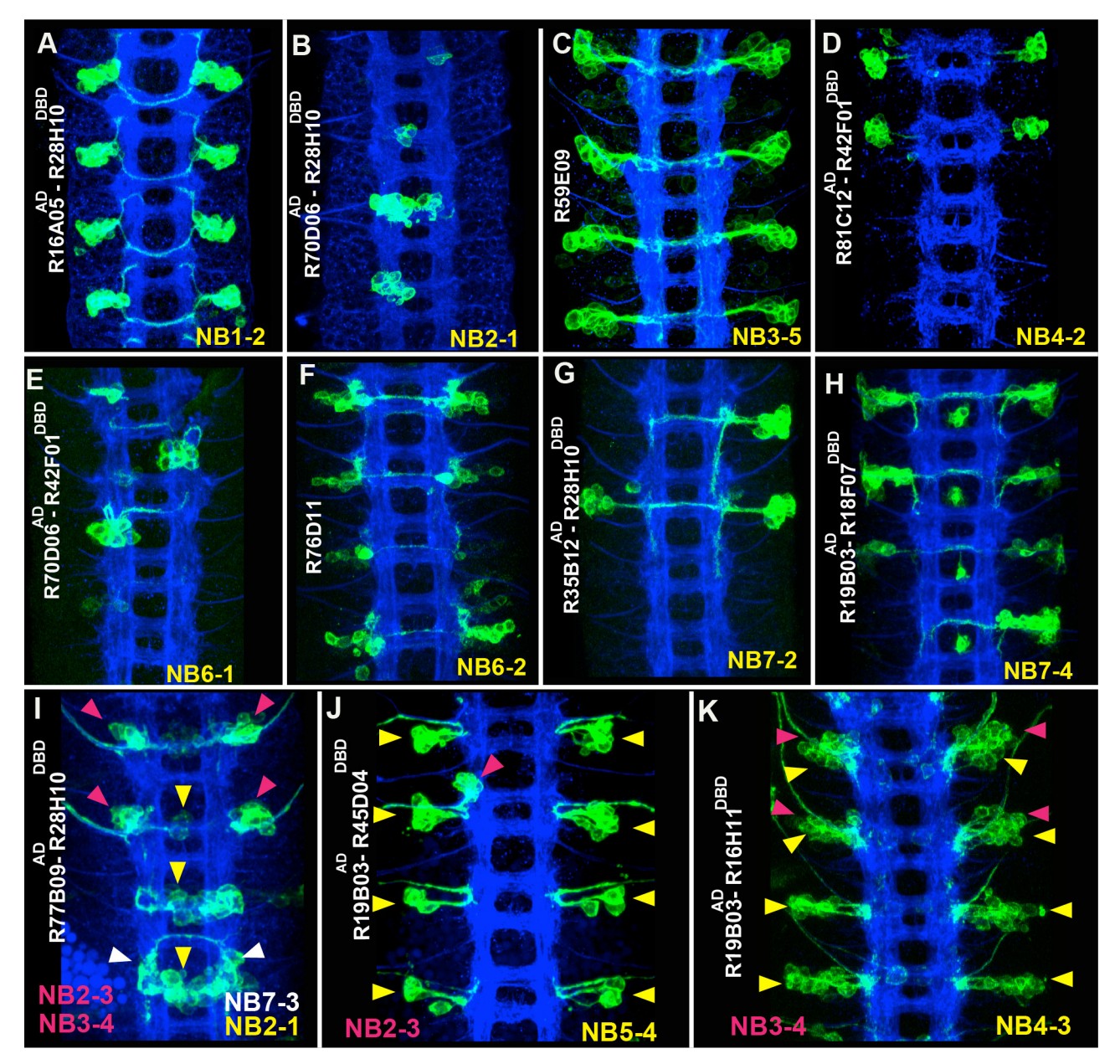

**Figure 2.** Sample GAL4 lines that mark NBs and their progeny. (A-K) Expression patterns of selected GAL4 lines in the nerve cords of late-stage embryos were visualized by driving mCD8-GFP (green). Only the T2-A2 segments are shown. (A-H) GAL4 lines uniquely label individual NBs and a subset of their progeny: NB1-2 (A), NB2-1 (B), NB3-5 (C), NB4-2 (D), NB6-1 (E), NB6-2 (F), NB7-2 (G), and NB7-4 (H). Although NB4-2 generates progeny in both thoracic and abdominal segments, R81C12$^{AD}$-R42F01$^{DBD}$ marks the NB4-2 lineage only in thoracic segments (D). (I-K) Expression of GAL4 lines that sparsely label a few NBs and their progeny. Color-coded arrowheads indicate the location of the NB lineages. See *Figures 2–4* for the presence of NBs, which are revealed by Dpn staining in some of these GAL4 lines. FasII+BP102 (blue) visualizes embryonic neuronal architecture; anterior is up.

coexpress in NB2-4 embryonically and in NB 18 postembryonically. Moreover, Unc-4 expression marks both the embryonic progeny of NB2-4 and the postembryonic neurons of lineage 18 (*Table 2*; *Lacin et al., 2014b*). Finally, lineage 18 is missing from T1 (*Truman et al., 2004*), and NB2-4 disappears from the anterior triplet by late embryogenesis (*Figure 3A,B*). Our conclusion that NB2-4 is the parent NB for lineage 18 differs from *Birkholz et al. (2015)*, who concluded NB3-4 generates lineage 18. The embryonic lineage clone that was used in their study to link NB3-4 to lineage 18

**Table 1.** GAL4 lines used to mark NBs and for reporter immortalization.

| Line | NB expression | Immortalization* | Figure |
|---|---|---|---|
| R16A05[AD]-R28H10[DBD] | NB1-2 | lin1 (30/30) | *Figure 2, Figure 4* |
| R70D06[AD]-R28H10[DBD] | NB2-1 | lin 2 (18/ 30), lin10 (7/ 30) | *Figure 2, Figure 4—figure supplement 1* |
| R19H09[AD]-R28H10[DBD] | NB2-2 | lin10 (16/30), lin7 (4/30) | *Figure 4—figure supplement 1* |
| R10C12 | NB2-3 | lin15 (30/30), lin25(7/30) | *Figure 5* |
| R65G02 | NB2-4 | lin 18 (21/30), lin20-22 (9/30), lin5 (5/30) | *Figure 3* |
| 5172J-Gal4 | NB3-1 | lin4 (12/18) | *Figure 4—figure supplement 1* |
| R21E09[AD]-R16H11[DBD] | NB3-2, NB4-2 | lin7 (18/24), lin13(8/24) | *Figure 4, Figure 4—figure supplement 1* |
| R21E09[AD]-R28H10[DBD] | NB1-2, NB3-2 | lin1 (15/24), lin7(8/24) | |
| ems-Gal4 | NB2-2, NB3-3, NB3-5 | lin10(10/30), lin8 (15/30), lin9 (20/30) | *Figure 3, Figure 4—figure supplement 1* |
| R59E09 | NB3-5 | lin9 (30/30) | *Figure 2, Figure 3* |
| R77B09[AD]-R28H10[DBD] | NB2-3, NB3-4, NB2-1 | lin15 (30/30), lin25(24/30), lin2 (20/30) | *Figure 1, Figure 4* |
| VT0048571 | NB4-1, NB7-2 | lin14 (34/36), lin11 (13/24) | *Figure 4—figure supplement 1* |
| R81C12[AD]-R42F01[DBD] | NB4-2 | Lin13 (30/30) | *Figure 2, Figure 4* |
| R19B03[AD]-R16H11[DBD] | NB4-3, NB3-4, NB2-3 | lin21 (30/30), lin25 (19/30), lin15 (8/30), | *Figure 2, Figure 4—figure supplement 1* |
| VT0041296 | NB4-4 | Lin24 (20/24); lin18 (5/18) | *Figure 4—figure supplement 1* |
| R54B10 | NB5-3, NB5-6 | lin5 (17/30) | *Figure 4—figure supplement 1* |
| R19B03[AD]-R45D04[DBD] | NB5-4, NB5-7, NB2-3 | lin20 (30/30), lin22(30/30), lin15 (10/30) | *Figure 2, Figure 6* |
| R24C10 | NB5-7 | lin20 (23/30), lin18(6/24) | *Figure 6* |
| R81F01 | NB6-1 | lin12 (13/30), lin18 (5/30) | |
| R70D06[AD]-R42F01[DBD] | NB6-1 | lin12 (12/24), lin13(7/24) | *Figure 2, Figure 4* |
| R76D11 | NB6-2 | lin19 (16/30) | *Figure 2; Figure 3* |
| R51B04 | NB7-1, NB6-2 | lin3 (22/30), lin19 (5/30) | *Figure 4—figure supplement 1* |
| R35B12[AD]-R28H10[DBD] | NB7-2 | lin11 (12/30) | *Figure 2, Figure 4* |
| R35B12 | NB7-1, NB6-2, NB7-2 | lin3(10/30), lin19 (6/30), lin11 (8/24) | |
| R19B03[AD]-R18F07[DBD] | NB7-4 | lin23 (26/30) | *Figure 2, Figure 4—figure supplement 1* |
| R19B03** | NB2-5, NB2-4 | lin17 (20/24), lin18 (26/30) | *Figure 3* |
| R13G03 | MNB | lin0 (10/18) | *Figure 4—figure supplement 1* |
| lbe-K-GAL4*** | NB5-6 | lin5-6 (20/20) | *Figure 8* |
| R45D04 | NB5-2, NB5-3, NB5-4, NB5-7, NB6-2 | lin6 (30/30), lin5(30/30), lin20 (30/30), lin22(30/30), lin19 (18/30) | |
| eg-GAL4 | NB2-4, NB3-3, NB3-4, NB6-4, NB7-3 | lin18 (19/30), lin8 (12/30) lin25(16/30) | *Figure 5, Figure 4—figure supplement 1* |

* The number of immortalized lineage per hemisegment shown in paranthesis. Corresponding NBs and lineages are color matched.

** Only dorsal part VNC scored

***NB5-6 generate postembryonic progeny only in S3 segments Lineages marked less than 15% of the time are not included. "lin" refers to postembryonic lineage.

**Table 2.** Expression profile of transcription factors in NBs and their corresponding embryonic and postembryonic progeny.

| NBs | lin | Embryonic progeny* | Postembryonic lin** | NB Marker*** |
|---|---|---|---|---|
| MNB | lin 0 | En, FoxD, Vg | En | En, unpg-Lacz |
| NB1-1 | lin16 | **Hb9, Lim3,** Isl, D, Eve, | Hb9, Lim3 | *mirr*-Lacz |
| NB1-2 | lin1 | **Msh, Nmr1,** Hb9, Nkx6 | Nmr1, Msh | *mirr*-Lacz |
| NB2-1 | lin2 | Toy | | *mirr*-Lacz |
| NB2-2 | lin10 | **Hb9, Lim3, Nkx6,** | Hb9, Lim3, Nkx6 | *mirr*-Lacz, Run |
| NB2-3 | lin15 | **Lim3, Nkx6, Isl,** | Isl, Lim3, Nkx6 | *mirr*-Lacz, Msh, Run |
| NB2-4 | lin18 | **Unc-4, Eg, Toy, Msh** | Unc-4 | *mirr*-Lacz, Msh |
| NB2-5 | lin17 | **Unc-4, Isl** | Unc-4, Isl | *mirr*-Lacz |
| NB3-1 | lin4 | **Hb9,** Msh | Hb9 | Nkx6, Run |
| NB3-2 | lin7 | Hb9, Toy, Barh, Unc-4 | Unc-4 | Ey, Dbx |
| NB3-3 | lin8 | **Toy, Lim3, Ems, Acj6,** Eg, Eve | Lim3, Ems, Acj6, Toy, Ey | Ems,Run |
| NB3-4 | lin25 | **Toy,** Ey, Msh, Eg | Toy, Nkx6 | Msh, Run, Ey, Eg-Gal4 |
| NB3-5 | lin9 | **Ems, Msh, Islet, Unc-4** | Ems, Msh, Islet | Ems |
| NB4-1 | lin14 | **Msh,** Unc-4 | Msh | *unpg*-Lacz |
| NB4-2 | lin13 | **Dbx, D, Vg,**Ey, Eve | Dbx, D, Vg | Ey |
| NB4-3 | lin21 | **Msh, Ey** | Msh, Ey | Ey, Msh |
| NB4-4 | lin24 | **Ems,** Toy | Ems, Toy | Ems, Ey, |
| NB5-1 | - | | | *gsb*-Lacz |
| NB5-2 | lin6 | **Toy, En, Vg,** Hb9 | Toy, En, Vg | *gsb*-Lacz, Run |
| NB5-3 | lin5 | **Vg, Toy,** Ey, En | Vg, Toy, | *gsb*-Lacz, Ey, Run |
| NB5-4 | lin22 | **BarH** | BarH | *gsb*-Lacz, Msh |
| NB5-5 | - | | | *unpg*-Lacz |
| NB5-6 | - | **EyA,** Toy | EyA (S3 segments) | *gsb*-Lacz,lbe-Gal4 |
| NB5-7 | lin20 | - | | *gsb*-Lacz, Msh |
| NB6-1 | lin12 | **Unc-4, Nmr1, Dbx** | Dbx, Unc-4, Nmr1 | *gsb*-Lacz, En, Dbx |
| NB6-2 | lin19 | **Unc-4, Dbx** | Dbx, Unc-4 | *gsb*-Lacz, En, D |
| NB6-4 | - | Eg, Toy, Msh | | *gsb*-Lacz, Eg-lacz |
| NB7-1 | lin3 | **Unc-4, Dbx, Eve** | Dbx, Nkx6, | *gsb*-Lacz, En |
| NB7-2 | lin11 | **Unc-4, Nkx6** | Unc-4, Nkx6, Eve | En, *unpg*-Lacz, Dbx |
| NB7-3 | - | Hb9, Isl, Eg, Ey | | En, *eg*-GAL4 |
| NB7-4 | lin23 | **Unc-4, Acj6** | Unc-4, Acj6 | En, Msh, D |

* Transcription factors that are also expressed in the postembryonic neurons are highlighted in bold.

** We failed to detect transcription factors in red in the corresponding embryonic progeny.

*** Expression of these markers is maintained from embryonic to postembryonic stages. "lin" refers to postembryonic lineage.

appears to be a partial NB2-4 clone that lacks early-born motor neurons. *Birkholz et al. (2015)* identified lineage 8 as the postembryonic progeny of NB2-4. However, our findings based on lineage tracings and molecular markers show that the neurons of lineage 8 arise from NB3-3 (see below).

We identified the posterior NB of the anterior triplet as NB3-5 based on its expression of Ems and lineage clones obtained from several independent GAL4 lines (*Figure 3F–H*; *Table 1*; *Moris-Sanz et al., 2014*). One of these lines, R59E09, uniquely marks NB3-5, and the immortalization of its expression labels lineage 9 in the larval VNC (30 of 30 hemisegments; *Figure 3I*). Moreover, we found that Ems expression is maintained in NB3-5 from embryonic to larval stages, and we also detected Msh, Islet (Isl) and Lim3 in both embryonic and postembryonic progeny of NB3-5 (*Table 2*; *Lacin et al., 2014b*).

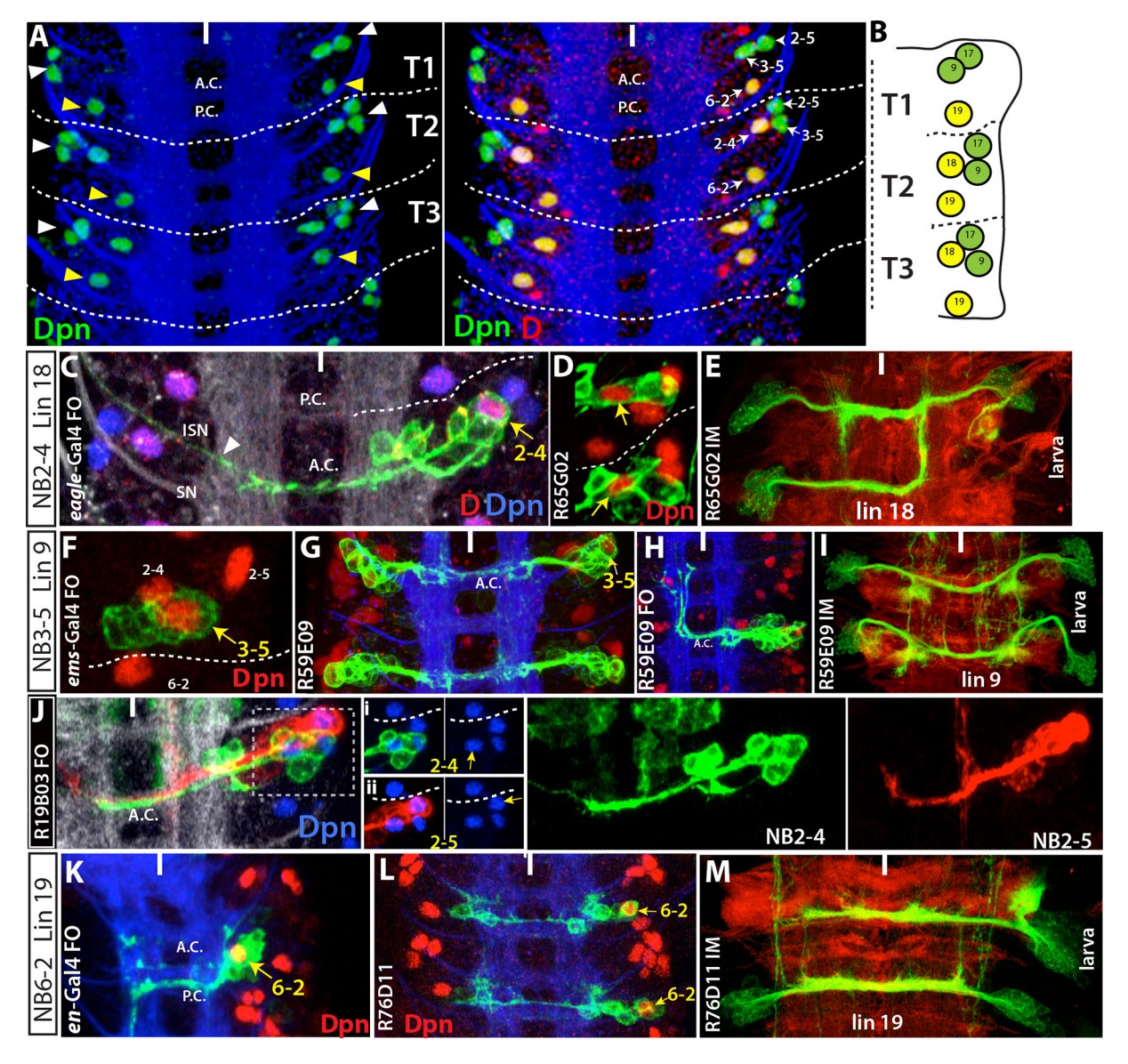

**Figure 3.** NB2-4, NB2-5, NB3-5 and NB6-2 generate lineage 18, 17, 9 and 19, respectively. Images on panels B, E, I, and M are from the larval CNS; the rest are from the embryonic CNS. (**A**) Projection of dorsally located NBs, marked by Dpn (green), in the thoracic segments of a stage-17 embryo. Three NBs (triplet) anteriorly and one NB posteriorly reside in each hemisegment (white and yellow arrowheads in the left panel, respectively). Note the T1 segment is missing a NB in the anterior region. Dichaete (red) marks the medial NB of the anterior triplet and the posterior NB (right panel). NB identities were determined based on lineage clones (see below). (**B**) Schematic view of the postembryonic lineages in the dorsal part of the larval thoracic nerve cord (depicted based on **Truman and Bate, 1988**, **Truman et al., 2004** and **2010**, and **Li et al., 2014**). Pattern of these postembryonic lineages is the same as that of dorsal embryonic NBs (compare **A** and **B**). (**C-E**) NB2-4 is the progenitor of lineage 18. (**C**) A NB2-4 lineage clone generated with *eg*-GAL4 includes the medial D[+] NB (**D**, red; Dpn, blue). A contralateral motor neuron crossing the midline via AC and exiting the CNS via ISN (arrowhead) is a characteristic feature of NB2-4 lineage. (**D**) R65G02 specifically marks NB2-4 and its progeny, which shows the same morphology as in (**C**) (not shown). (**E**) Immortalization of R65G02 in NB2-4 progeny constantly and specifically marks lineage 18 in the larval CNS. (**F-I**) NB3-5 is the progenitor of lineage 9. (**F**) *ems*-GAL4 marks NB3-5 which is the posterior NB in the triplet. (**G**) R59E09 specifically labels NB3-5 and its progeny. (**H**) A NB3-5 lineage clone obtained with R59E09. (**I**) Immortalization of R59E09 uniquely marks lineage 9. (**J**) R19B03 marks NB2-4 (green clone) and NB2-5 (red clone) lineages in the embryonic CNS. NB2-5 is located anteriorly (**J-i**) and NB2-4 is located medially (**J-ii**). (**K-M**) NB6-2 is the progenitor of lineage 19. (**K**) The posterior NB is located in a NB6-2 lineage clone obtained with *en*-GAL4. (**L**) R76D11 specifically marks NB6-2 and (**M**) its immortalization in NB6-2 progeny visualizes lineage 19. Wavy dashed lines indicate segment boundaries. White bar indicates the midline. FO, flip-out lineage clone; IM, immortalization; A.C., anterior commissure; P.C., posterior commissure; SN, segmental nerve; ISN, inter segmental nerve. In this and subsequent figures, lineage is abbreviated as "lin".

The remaining NB in the anterior triplet is NB2-5. As shown in *Figure 3J*, R19B03 (a driver from the Msh locus) labels two NBs with their progeny in the dorsal nerve cord: one is NB2-4 and the other is NB2-5 based on its distinctive intersegmental neurons, which extend ipsilaterally into the anterior segments. Reporter immortalization of R19B03 labels cells of both lineage 17 and lineage 18 in the dorsal nerve cord (not shown and *Table 1*). Thus, NB2-5 generates lineage 17 since NB2-4 generates lineage 18.

Flip-out clones generated by *engrailed (en)*-GAL4 contained the most posterior NB in the dorsal part of the nerve cord. Neurons in these clones extend two separate bundles across the posterior commissure and express Dbx (*Figure 3K* and not shown). These are unique features of NB6-2 (*Schmidt et al., 1997*; *1999*; *Lacin et al., 2009*), and thus we identified this progenitor cell as NB6-2. R76D11 labels NB6-2, and reporter immortalization of R76D11 marks lineage 19 (16 of 30 hemi-segments; *Figure 3L,M*). The dorsal location of NB6-2 in late-stage embryos (*Figure 3K*) compared to its ventral position in early-stage embryos (*Doe, 1992*) suggests that this cell migrates dorsally during development. Indeed, using the R76D11 reporter, we detected NB6-2 in a ventral position in stage-12 embryos, becoming more dorsal as the embryos aged. In both embryonic and larval nerve cords, the NB of this lineage expresses *gsb*-LacZ, En, and D, and the embryonic and postembryonic neurons express Dbx and Unc-4 (*Table 2*; *Lacin et al., 2014b*).

## A lineage map of all NBs in the VNC

Using the same approaches as detailed for the dorsally located NBs, we unambiguously identified the postembryonic progeny of all the NBs except NB1-1 and NB5-2, for which we failed to identify specific GAL4 lines (*Figure 4—figure supplement 1*). To identify the postembryonic progeny of these last two NBs, we used a fact that emerged during our lineage-tracing experiments: in almost all lineages, embryonic and postembryonic progeny of a NB showed significant similarities in how they extended their axons and in the molecular markers they expressed. For example, a subset of the embryonic progeny of NB1-2 send axons ipsilaterally to the next anterior segment while others extend contralateral axons that turn with a characteristic posterior hook (*Figure 4Aii*). The same pattern is seen in the postembryonic progeny of this NB, with the set of 1B interneurons projecting anteriorly into the next ipsilateral neuropil and their 1A sibs crossing the midline and showing the posterior hook (*Figure 4Aiii*; *Truman et al., 2004*). We used this strategy of comparing embryonic and postembryonic projections and molecular markers to identify the progeny of NB1-1 and NB5-2.

In stage-16 embryos, NB1-1 is located medial to NB1-2 (not shown). Since NBs appear to retain their relative positions between stage-16 embryos and larval stages (e.g., dorsal NBs in *Figure 3*), we expected that the postembryonic progeny of NB1-1 would lie medial to NB1-2 progeny (lineage 1). Lineage 16 resides medial to lineage 1 (*Truman et al., 2004*), and in agreement with this, we found that embryonic interneurons from NB1-1 express Hb9 and Lim3 and extend their axons laterally in a similar manner to lineage 16 postembryonic interneurons that are also Hb9$^+$ Lim3$^+$ (*Table 2*; *Figure 4—figure supplement 2A*). Moreover, using NB-specific GAL4 lines we had assigned all of the surrounding NBs to other postembryonic lineages. Thus, we concluded that NB1-1 generates lineage 16. With a similar strategy we identified lineage 6 as the postembryonic progeny of NB5-2. We found that both NB5-2 embryonic progeny and lineage 6 neurons express En, Twin of eyeless (Toy), and Vestigial (Vg), and their axons use the same routes in the posterior commissures to cross the midline (*Table 2*; *Figure 4—figure supplement 2B*).

We failed to find postembryonic progeny of NB5-1, NB5-5, NB5-6, NB6-4, or NB7-3, suggesting these NBs lack the second, postembryonic neurogenic phase. Indeed, NB5-6 is eliminated by apoptosis during embryogenesis (*Baumgardt et al., 2009*), and a similar fate was concluded for NB7-3 (*Karcavich and Doe, 2005*). To verify these findings and determine the fates of the other NBs, we surveyed the VNC by TUNEL staining, which marks dying cells. In the thoracic segments, NB7-3, NB6-4 and NB5-6 were TUNEL-positive in stage-14, -15 and -16 embryos, respectively, and could no longer be found in stage-17 embryos (*Figure 4—figure supplement 3C–G*; not shown). Hence, these three NBs die late in embryogenesis. Further, while we detected NB5-1 and NB5-5 in stage-13 embryos by *gsb*-lacZ and/or *unplugged (unpg)*-lacZ expression, we could not identify them at stage 17 (*Figure 4—figure supplement 3A* and not shown). However, we did not observe any TUNEL staining in either NB. Either they died and we missed their TUNEL-positive window, or they lost their

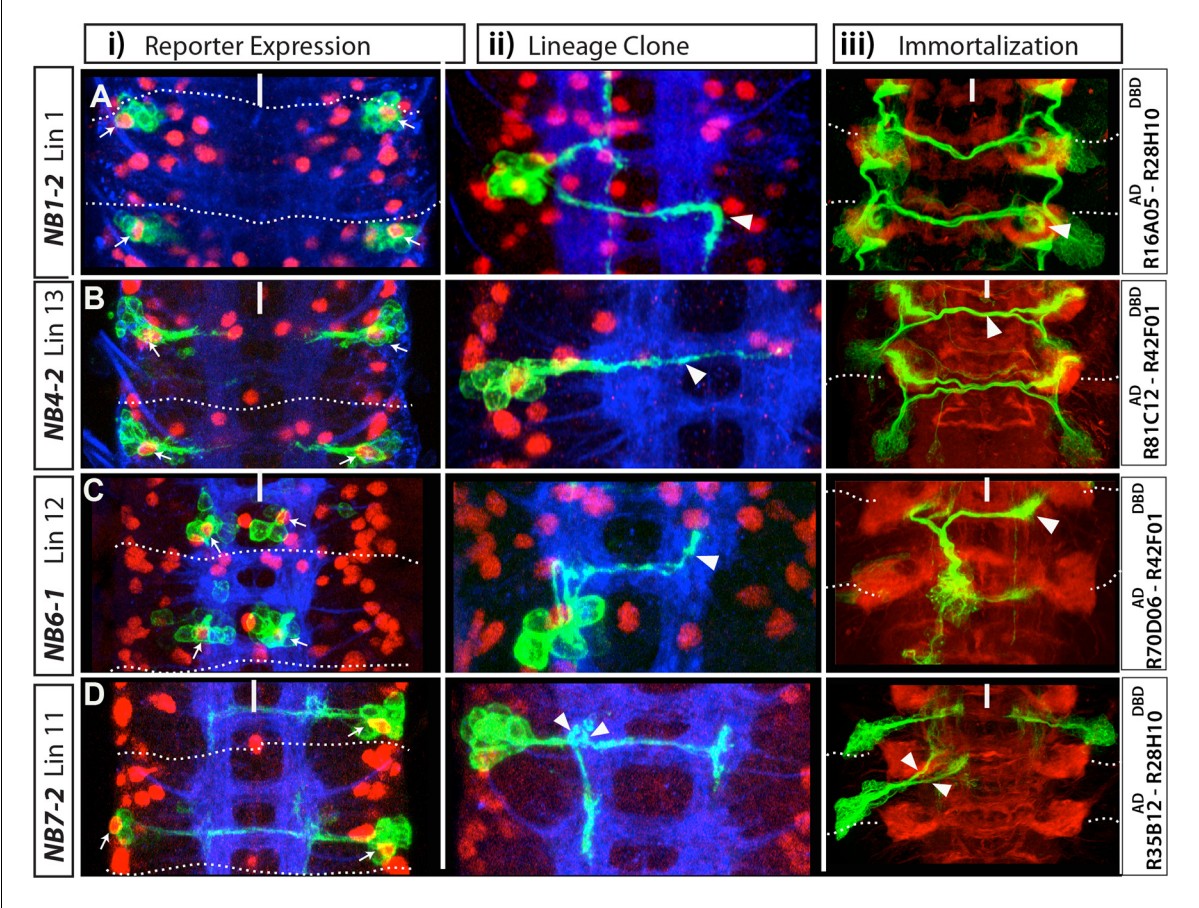

**Figure 4.** NB1-2, NB4-2, NB6-1 and NB7-2 generate lineages 1, 13, 12 and 11, respectively. (**A-D**) Four examples showing identification of postembryonic lineages via reporter immortalization of NB-specific GAL4 lines. (**i** and **ii**) Nerve cords dissected from late stage embryos; Dpn (red) marks NBs. (**i**) Indicated GAL4 lines drive GFP reporter expression in an individual NB (arrows) in each hemisegment in addition to some of its progeny. **ii**) Flip-out lineage clones showing the morphology of individual embryonic NB lineages were used to find the identity of NBs. (**iii**) Nerve cords from wandering stage larvae in which reporter expression of the indicated GAL4 lines was immortalized in the progeny of NBs. This technique almost exclusively marks postembryonic progeny and rarely embryonic progeny (see Materials and methods and *Figure 1*). In each hemisegment, reporter immortalization of NB specific GAL4 lines marks a single postembryonic lineage. Arrowheads (**ii** and **iii**) show the similarities in neuronal morphology between embryonic and postembryonic progeny of an individual NB. For example, contralateral axons of lineage 1 turn with a characteristic posterior hook; embryonic neurons of the same lineage also exhibit a similar turn (**Aii-iii**). NB7-2 postembryonic progeny (lineage 11) extend two ipsilateral axonal bundles; NB7-2 embryonic neurons have similar axonal projections (arrows in **Dii-iii**). Dashed lines indicate segment boundaries. White bar indicates the midline. HRP (blue) in embryos (**i, ii**) and Phalloidin (Red) in larvae (**iii**) visualize neuronal architecture.

The following figure supplements are available for figure 4:

**Figure supplement 1.** Postembryonic progeny of the remaining NBs.

**Figure supplement 2.** (A) NB1-1 generates lineage 1.

**Figure supplement 3.** (A-B) NB5-5 disappears in the late stage embryos.

progenitor and underwent terminal differentiation as is seen in some of the other NBs at the end of their postembryonic phase (*Maurange et al., 2008*).

The lineage map that we determined based on the above approaches is shown in *Figure 1B* and differs for eight of the lineages from the map by *Birkholz et al., (2015)* based on the flybow approach (*Table 3*). We differ on the designations for lineages 7, 8, 11, 13, 15, 18, 20, and 25. We

**Table 3.** Lineages that are assigned to different NBs by this study and *Birkholz et al. (2015)*.

| Lin | Findings on lineage tracing supporting assignments of this study | Findings on molecular markers supporting assignments of this study |
|---|---|---|
| lin7<br>NB3-2*<br>NB4-2** | Immortalization of R21E09$^{AD}$-R28H10$^{DBD}$, which marks lin7, labels NB3-2 in the embryo, but not NB4-2. | NB3-2 and NB of lin7 are Dbx$^+$ versus NB4-2 is Dbx$^-$. Embryonic progeny of NB3-2 and lin7 neurons are Unc4$^+$ versus NB4-2 progeny are Unc4$^-$. |
| lin8<br>NB3-3*<br>NB2-4** | Immortalization of *ems*-GAL4, which marks lin8, labels NB3-3 in the embryo, but not NB2-4 (*Figure 4—figure supplement 1*). | NB3-3 and NB of lin8 are Ems$^+$ versus NB2-4 is Ems$^-$. Embryonic progeny of NB3-3 and lin8 neurons are Acj6$^+$ versus NB2-4 progeny are Acj6$^-$. |
| lin11<br>NB7-2*<br>NB6-4** | Immortalization of R35B12$^{AD}$-R28H10$^{DBD}$, which specifically marks lin11, labels only NB7-2 in the embryo, but not NB6-4 (*Figure 4*). NB6-4 dies (TUNEL+) in late stage embryos (*Figure 4—figure supplement 3*). | NB7-2 and NB of lin11 are marked by *unpg*-LacZ versus NB6-4 is not. NB6-4 is Msh$^+$ versus NB of lin11 is Msh$^-$. Embryonic progeny of NB7-2 and lin11 neurons are Unc4$^+$ versus NB6-4 progeny are Unc4$^-$. |
| lin13<br>NB4-2*<br>NB3-3** | Immortalization of R81C12$^{AD}$-R42F01$^{DBD}$, which specifically marks lin13, labels only NB4-2 in the embryo, but not NB3-3 (*Figure 4*). | NB4-2 and NB of lin13 are Ey$^+$ versus NB3-3 is Ey$^-$. Embryonic progeny of NB4-2 and lin13 neurons are Dbx$^+$/D$^+$/Vg$^+$ versus NB3-3 progeny are Dbx$^-$/D$^-$/Vg$^-$ (*Figure 11*). |
| lin15<br>NB2-3*<br>NB3-2** | Immortalization of R10C12, which specifically marks lin15, labels only NB2-3 in the embryo, but not NB3-2 (*Figure 5*). | Lineage 15 NB and NB2-3 are Msh$^+$ versus NB3-2 is Msh$^-$. |
| lin18<br>NB2-4*<br>NB3-4** | Immortalization of R65G02, which specifically marks lin18, labels only NB2-3 in the embryo, but not NB3-4 (*Figure 5*). Dorsal location of NB2-4 correlates with lineage 18. | Embryonic progeny of NB2-4 and lin18 neurons are Unc4$^+$ versus NB3-4 progeny are Unc4$^-$. |
| lin20<br>NB5-7*<br>NB5-4** | Immortalization of R24C10, which marks lin20, labels NB5-7 in the embryo, but not NB5-4 (*Figure 6*). | NB5-7 shares similar molecular markers with NB5-4. |
| lin25<br>NB3-4* -** | Lineage tracing via *eg*-GAL4, R77B09$^{AD}$-R28H10$^{DBD}$, and R19B03$^{AD}$-R16H11$^{DBD}$ lines, all of which mark lin25, label NB3-4 in the embryo (*Figure 5*). | Lineage 25 is first described here and assigned to NB3-4, which has not been previously characterized in detail. |

*NB assignment by this study

** NB assignment by *Birkholz et al. (2015)* "lin" refers to postembryonic lineage

identified NBs 2–3, 3–4, and 5–7 as the stem cells for lineages 15, 25, and 20, respectively. NBs 2–3 and 3–4 had not been characterized prior to this study, and NB5-7 was first described here, thus identification of these NBs and their progeny is explained in detail below (*Figures 5*, *6*). We found NB3-2, NB3-3, NB7-2, NB 4–2 and NB2-4 were the stem cells for lineages 7, 8, 11, 13 and 18, respectively (*Figure 3C*, *4B,D* and *Figure 4—figure supplement 1D, L*).

## Postembryonic lineages 7 and 13 are produced by NB3-2 and NB4-2, respectively

NB3-2 and NB4-2 generate progeny with similar morphologies: both generate ipsilateral motor neurons and contralateral interneurons. The exit routes of their motor neurons from the CNS, however, are different (*Bossing et al., 1996*; *Landgraf et al., 1997*; *Schmid et al., 1999*). We screened for molecular markers that would unambiguously identify these lineages in the embryo. We found that NB3-2 expresses Dbx, and its interneurons express Unc-4 whereas NB4-2 is negative for Dbx, and its interneurons express D and Dbx (*Table 2*). Based on this information, we identified the split-GAL4 combination R81C12$^{AD}$-R42F01$^{DBD}$, which uniquely marks NB4-2 in the embryo. Immortalization of this reporter line marks lineage 13 (30 out of 30 hemisegments), thus indicating NB4-2 generates lineage 13. In agreement, lineage 13 neurons in the larva express D and Dbx (*Lacin et al., 2014b*).

R21E09$^{AD}$-R16H11$^{DBD}$ marks NB3-2 consistently and NB4-2 occasionally. (*Figure 4—figure supplement 1D*). Its reporter immortalization marks lineage 7 (18 out of 24 hemisegments) and lineage 13 (8 out of 24 hemisegments). We concluded that NB3-2 generates lineage 7, since NB4-2 generates lineage 13. In support of this conclusion, we found that Dbx marks both NB3-2 and the NB of lineage 7 and Unc-4 marks both NB3-2 embryonic progeny and postembryonic lin7 neurons (*Table 2*; *Lacin et al., 2014b*). Moreover, immortalization of another driver combination, R21E09$^{AD}$-R28H10$^{DBD}$, which marks NB3-2 in the embryo, identifies lineage 7 as its progeny (*Table 1*).

## Postembryonic lineage 8 is produced by NB3-3

*ems*-GAL4 marks NB2-2, NB3-3, and NB3-5 and its immortalization labels lineages 10, 8, and 9 (*Figure 4—figure supplement 1L*; *Estacio-Gomez et al., 2013*; *Moris-Sanz et al., 2014*). Lineage tracing via NB-specific GAL4 lines identified postembryonic lineages 10 and 9 as progeny of NB2-2 and NB3-5, respectively (*Figure 4—figure supplement 1B*; *Figure 3F–I*), thus leaving the lineage 8 neurons as the progeny of NB3-3. This conclusion is supported by our detection of the Ems protein, which marks embryonic NB3-3 (*Hartmann et al., 2000*), in the NB of lineage 8 and the presence of common transcription factors in both embryonic progeny of NB3-3 and postembryonic lineage 8 neurons (*Table 2*). In addition, the axonal morphology of thoracic NB3-3 embryonic progeny is virtually identical to the morphology of postembryonic lineage 8 neurons (*Figure 4—figure supplement 1L*). Thus, we concluded NB3-3 generates lineage 8 neurons. This designation differs from *Birkholz et al. (2015)* who concluded that NB2-4 was the progenitor of lineage 8. We think their designation is unlikely since NB2-4 is located in the dorsal surface of the nerve cord (unlike NB3-3 and lineage 8), does not express Ems and generates the dorsally located lineage 18 (*Figure 3C–E*).

## Postembryonic lineage 11 is produced by NB7-2

Reporter immortalization of R35B12$^{AD}$-R28H10$^{DBD}$ marks specifically lineage 11 (12 of 30 hemisegments; *Figure 4D*). In the embryo, R35B12$^{AD}$-R28H10$^{DBD}$ marks a single NB. Progeny of this NB have contralateral projections across the posterior commissure and intersegmental ipsilateral projections extending posteriorly. These are the unique features of the NB7-2 lineage. Thus, we concluded that NB7-2 generates lineage 11. *Birkolz et al. (2015)* identified NB6-4 as the stem cell for lineage 11 based on embryonic flybowl clones (*Birkholz et al., 2015*). Although the embryonic progeny of NB6-4 show morphological similarities to the NB7-2 lineage, the ipsilateral projections of NB6-4 lineage are short and local. Our findings, which indicate that NB6-4 is eliminated by apoptosis during embryogenesis, also favor NB7-2 as the founder cell of lineage 11. Moreover, *unpg*-lacZ, which labels NB7-2 but not NB6-4 (*Doe, 1992*) marks the NB of lineage 11 (*Figure 4—figure supplement 3H*).

## Postembryonic lineage 15 is produced by NB2-3

In the embryo, R10C12-GAL4 marks a single NB in each thoracic hemi-segment (*Figure 5A*). We immortalized R10C12 expression in its progeny and found that this NB generates lineage 15, which is composed of a large number of leg motor neurons and glial cells (*Figure 5C*). To find the identity of this NB, we assayed a panel of molecular markers and found that *mirror*-lacZ, Runt, and Msh label this NB (*Figure 5A*; not shown). Based on previous studies, NB2-3 is the only NB that delaminates from Msh$^+$ neuroectoderm and expresses mirror-lacZ and Runt (*Doe, 1992*; *Broadus et al, 1995*; *Dormand and Brand, 1998*; *Isshiki et al., 1997*). Also, the NB marked by R10C12 typically resided in row two and column three of the NB layer (Supp. *Figure 5A*). Little is known about NB2-3 and its progeny, but it was previously suggested that it produced leg motor neurons (*Schmid et al., 1999*). Two previous studies attempted to find embryonic progeny of NB2-3 via DiI labeling; both experienced difficulties and observed unusual rates of cell death (*Schmidt et al, 1997*; *1999*). One study was able to label NB2-3 progeny only in the thoracic segments and showed that NB2-3 progeny comprise a few extraordinarily large cells without axons (9 microns in diameter) and a few interneurons with contralateral projections (*Schmid et al., 1999*). To visualize NB2-3 embryonic progeny with a less invasive approach, we generated lineage clones with R10C12 and three additional independent lines (R11B05, R28H10 and R77B09$^{AD}$-R28H10$^{DBD}$), all of which label NB2-3. We recovered lineage clones containing this NB only from the thoracic segments (n>50). The largest clones, presumably the entire embryonic lineage, contained 10 cells in addition to the NB (n=8), and the only axonal projection was into the periphery unlike the previously documented NB2-3 progeny (*Schmid et al., 1999*). These axons exited the CNS in two bundles via the segmental nerve but did not innervate any muscles in stage-17 embryos (*Figure 5B*). We obtained similar-looking clones with pan-neuronal *elav*-GAL4, indicating that embryonic progeny of NB2-3 extend only efferent axons (n=3; not shown). Like postembryonic lineage 15 neurons, NB2-3 embryonic progeny express Lim3, Isl and Nkx6. *Birkholz et al. (2015)* identified NB3-2 as the progenitor of lineage 15, but NB3-2 does not express Msh, whereas NB2-3 in the embryo and lineage 15 NB in the larva do both express Msh (*Figure 5A, L*).

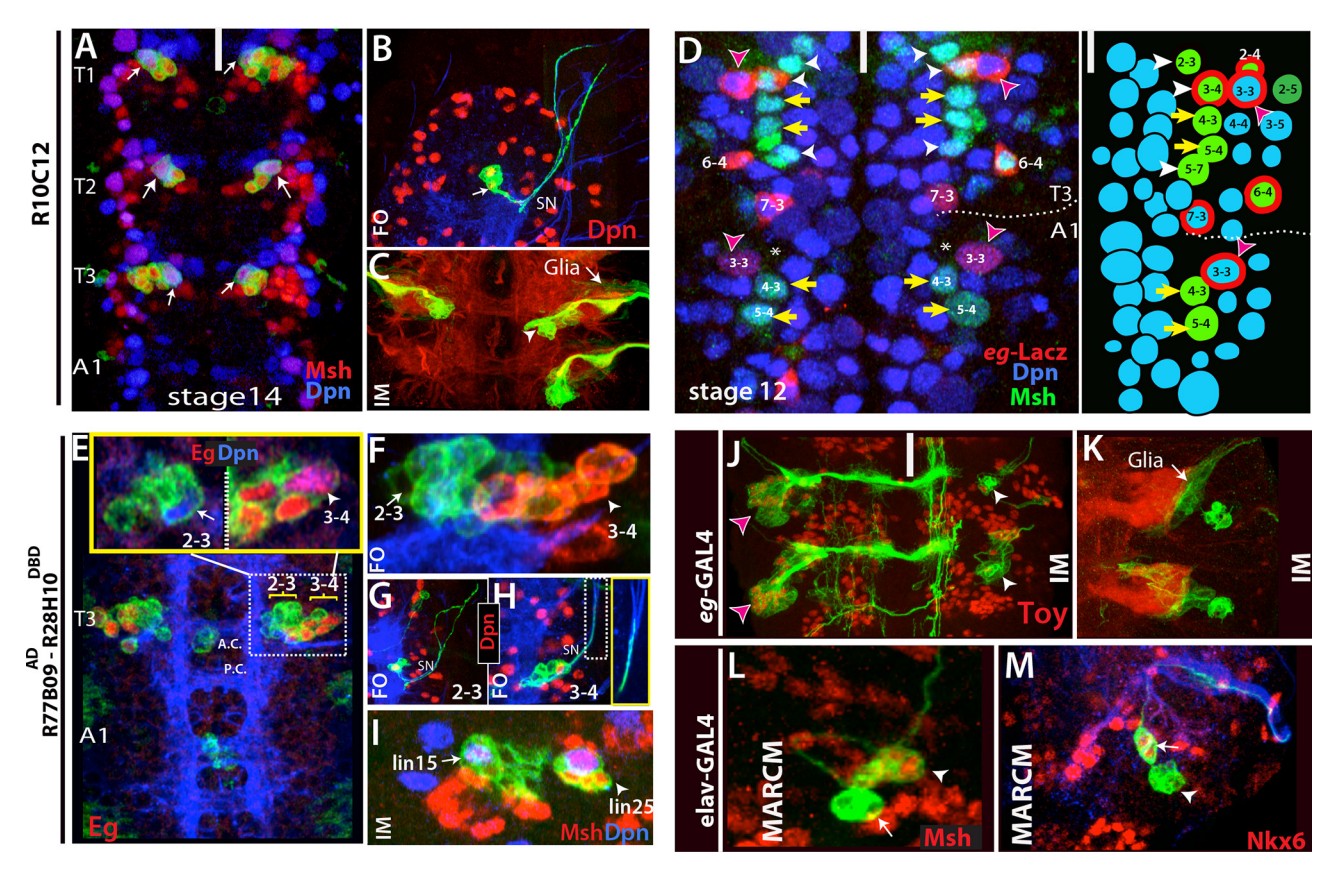

**Figure 5.** Thoracic-specific NB2-3 and NB3-4 generate lineage 15 and lineage 25, respectively. (**A**) R10C12 marks uniquely Msh[+] NB2-3 (arrows) in each thoracic hemisegment but not in abdominal ganglia. Dpn, blue; Msh, red (**B**) A lineage clone containing NB2-3 (arrow) extends only efferent axons, which splits into two bundles after exiting the CNS via the segmental nerve (SN). (**C**) Immortalization of R10C12 expression in the NB2-3 progeny marks lineage 15, which is composed of motor neurons (arrowhead) and glia (arrow). (**D**) T3 and A1 segments from a stage 12 embryo shown. *eg*-lacZ (red) and Msh (green) co-staining reveal differences in NB (blue) pattern of T3 and A1 segments. Schematic representation is shown on the right. Msh marks three thoracic specific NBs (NB2-3, NB3-4, and NB5-7; white arrowheads). NB3-4, which is marked by both *eg*-lacZ and Msh, resides medial to NB3-3. In abdominal segments, no NB with lacZ and Msh expression is detected medial to NB3-3; note the region immediately medial to NB3-3 does not have any NB (asterisks). (**E-I**) R77B09[AD]-R28H10[DBD] used to identify the postembryonic progeny of NB2-3 and NB3-4. (**E-G**) Nerve cords from late stage embryos shown. NB3-4 (arrowhead) and its progeny express Eg (red) while NB2-3 (arrow) and its progeny do not. Inset shows the magnified views of NB2-3 (left) and NB3-4 (right) lineages. (**F**) A lineage clone via R77B09[AD]-R28H10[DBD] separates two NB lineages: NB3-4 lineage is marked with red. Progeny of NB3-4 (**H**) extend axons out of the CNS using the SN route like NB2-3 progeny (**G**) but do not extend them as far as NB2-3 progeny (see inset **H**). (**I**) Nerve cord from a late-stage larva shown. Msh, red; Dpn, blue. Immortalization of R77B09[AD]-R28H10[DBD] expression marks lineage 15 (NB2-3 progeny) and lineage 25 (NB3-4 progeny). (**J**) Immortalization of *eg*-GAL4 marks Toy expressing lineage 8 (NB3-3 progeny; red arrowheads; see also *Figure 4—figure supplement 1L*) and lineage 25 (white arrowheads). (**K**) Lineage 25 contains glial cells (arrow). (**L-M**) *elav*-MARCM clones showing lineage 25 have a small cluster of Msh expressing cells (arrowheads) and two Nkx6 expressing motor neurons (arrows). Msh expressing cells reside closely to Msh[+] NB3-4, but do not extend any processes (arrowheads in **L** and **M**). Note glial cells in lineage 25 are not visible in MARCM clones since *elav*-GAL4 does not mark glial cells. White bar indicates the midline. FO, flip-out lineage clone; IM, immortalization; A.C., anterior commissure; P.C., posterior commissure; white bar, midline.

The following figure supplement is available for figure 5:

**Figure supplement 1.** (**A**) Stage 12 embryonic nerve cord; Engrailed (En; red) marks row 6 and 7 NBs as well as NB1-2.

## Postembryonic lineage 25 is generated by NB3-4

We found a new postembryonic lineage (named lineage 25) that was previously overlooked, presumably because of its small number of progeny. We observed lineage 25 when we immortalized reporter expression of R19B03[AD]-R16H11[DBD], *eg*-GAL4, and R77B09[AD]-R28H10[DBD] lines (*Figure 5I,*

K; not shown). It also appeared in MARCM clones generated with *elav*-GAL4 and *actin*-GAL4 lines (*Figure 5L,M*; not shown). At the wandering larval stage, lineage 25 is located lateral to lineage 15 (NB2-3), and includes the NB, a small cluster of closely adhering cells expressing Msh, and two motor neurons expressing Nkx6 (*Figure 5I,L, and M*). It is similar to lineage 15 in several ways: (i) neurons in both postembryonic lineages extend their axons towards leg discs, indicating they are leg motor neurons (not shown); (ii) both lineages contain glial cells located around the leg neuropil (*Figure 5C,K*); and (iii) both NBs express Msh and Runt (*Figure 5I*; not shown). To find the embryonic origin of lineage 25, we generated lineage clones in the embryo with the lines mentioned above. Common to all lines, we identified a thoracic specific NB, which is located ventrolaterally. For example, reporter immortalization of R77B09$^{AD}$-R28H10$^{DBD}$ results in lineage 15 and more laterally lineage 25 in addition to medially residing lineage 2 neurons (*Figure 5E,I*; lineage 2 not shown). Based on the lineage clones generated with this line in the embryonic CNS, we found that R77B09$^{AD}$-R28H10$^{DBD}$ marks NB2-1 (the stem cell for lineage 2; not shown), NB2-3 (stem cell for lineage 15; *Figure 5E,F*), and a more lateral NB (the stem cell for lineage 25; *Figure 5E,G*). The lateral NB had five progeny in their lineage clones and their axons exit the CNS in the segmental nerve but do not extend as far as the NB2-3 motor neurons (*Figure 5G*). This NB and its progeny in the embryo expressed Msh, Runt, Eg, and Eyless (Ey) (*Figure 5D,E*; not shown), and it was the only NB found in common to all of the GAL4 lines whose reporter immortalization marks lineage 25 (*Figures 5I,7J-K*; not shown). Hence, we concluded that it is the stem cell of lineage 25.

The embryonic designation of this stem cell was difficult to determine because no NB expressing Msh, Runt, Eg, and Ey and giving rise to progeny with this morphology, has been reported (*Schmidt et al., 1997*; *1999*; *Birkholz et al., 2013*). Among these transcription factors, Eg has been used in many studies because it marks only four NBs: NB2-4, NB3-3, NB6-4, and NB7-3 (*Higashijima et al, 1996*; *Dittrich et al., 1997*; *Mettler et al. 2006*, *Tsuji et al., 2008*). However, when we analyzed *eg*-lacZ, eg-Gal4, or Eg protein expression in the nerve cords of stage-12 embryos, when NB delamination was complete, we observed four Eg$^+$ NBs in abdominal segments, but five such NBs in the thoracic segments (*Figure 5—figure supplement 1B-C*). The extra Eg$^+$ NB in the thorax is the lineage 25 stem cell, since it is also labeled by Msh, Runt, and Ey (*Figure 5D*; not shown). This NB resides in the third row and third column of the thoracic NB array. This position is perplexing because we find NB3-3, which is characterized by being Msh$^-$ and Eg$^+$, in the fourth position in the third row of the thoracic array, although it occupies the third position in the abdominal array (*Figure 5D*). Although misplaced, we propose this thoracic-specific NB is NB3-4 because: (i) NB3-4 is the only NB in the vicinity that has not been well characterized, and previous studies failed to identify its progeny with confidence (*Schmidt et al., 1997*; *1999*); and (ii) in both abdominal and thoracic segments, only three NBs reside near but lateral to NB3-3: the Msh$^+$ NBs 2–4 and NB2-5, and the EMS$^+$ NB3-5, indicating that NB3-4 cannot be lateral to NB3-3. Earlier studies may have confused NB3-3 and NB3-4 (*Doe, 1992*; *Higashijima et al., 1996*). In agreement with our findings, a recent study in *T. castaneum* identified an Msh$^+$ NB in the third row and third column of the NB array in the thoracic segments. However, the authors identified this NB as NB3-3 and explained the discrepancy in Msh expression as a difference between two species (*Biffar and Stollwerk, 2015*). Since lineage 25 is first described here, *Birkholz et al. (2015)* did not attempt to identify its stem cell; however, they assigned NB3-4 as the progenitor of lineage 18 based on a Flybow lineage clone that appears to be a partial NB2-4 clone (*Birkholz et al., 2015*). As discussed above, the molecular markers argue that lineage 18 is produced by NB2-4, and we conclude that NB3-4 gives rise to the newly described lineage 25.

## Lineage 20 is generated by NB5-7

Lineage 20 and lineage 22 are postembryonic lineages with very similar morphological and molecular features. Axon bundles of both lineages extend similarly and apparently terminate in neighboring compartments of the leg neuropil, NBs of both lineages express Msh, and neurons in both lineages express BarH (*Truman et al, 2004*; *Lacin et al., 2014b*). Reporter immortalization of R19B05$^{AD}$-R45D04$^{DBD}$ marks both of these lineages in the larval CNS (*Figure 6G*). In the embryonic nerve cord, this driver marks two neighboring NBs, presumably the progenitors of these two lineages (*Figure 6A–F*). Both NBs express Msh and *gsb*-lacZ, and neither of them expresses En (which marks row 6/7 NBs) or *unpg*-lacZ (which marks NB5-5) (*Figure 6C-D, E*; not shown). Previous studies

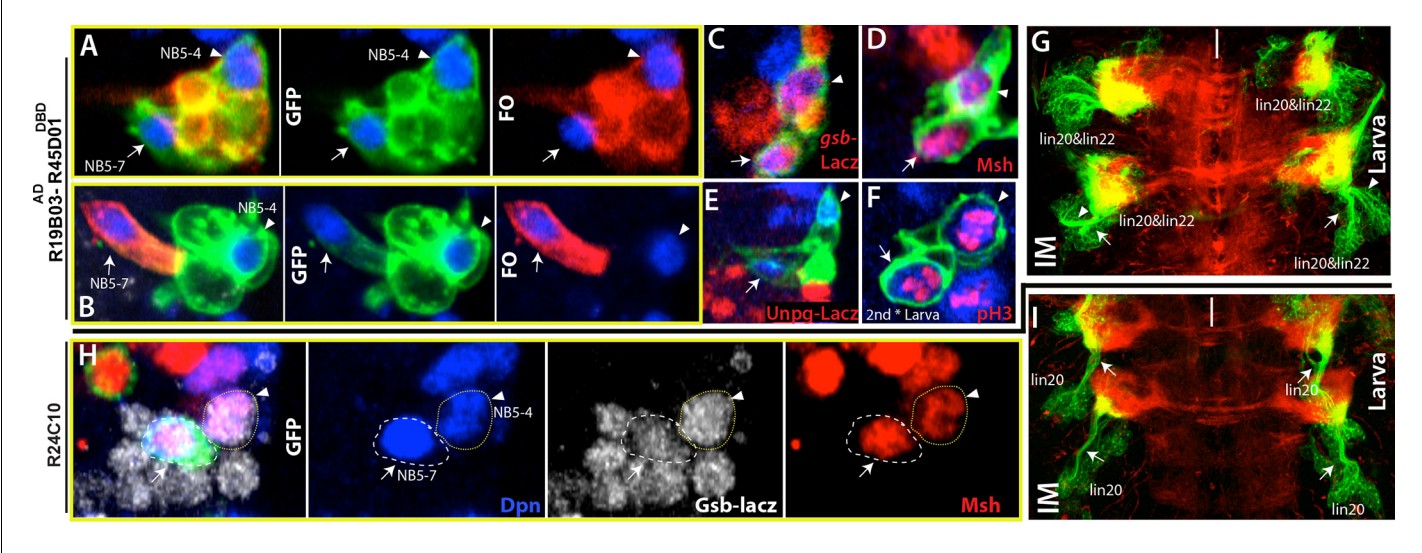

**Figure 6.** NB5-4 and NB5-7, a novel NB, generate almost identical lineages, lineage 22 and 20, respectively. (A, F) R19B03[AD]-R45D04[DBD] marks two adjacent NB in the embryonic (A-E) and the larval CNS (F). Hemisegments on the right of the midline are shown. (A-B) GFP (green) shows complete GAL4 expression pattern; lineage clones are in red. (A) NB5-4 (arrowheads) resides usually anterior and lateral to the other NB (arrows) and contains several motor neurons in its progeny (red clone). (B) The other NB (arrow), named NB5-7, does not have any progeny (red cell) during embryogenesis. (C-E) Molecular characterization of NB5-4 and NB5-7: *gsb*-lacZ (C) and Msh (D) co-label both NBs and *unpg*-lacZ, which marks NB5-5 dorsally (not shown), spares both NBs. (F) pH3, a mitotic marker, labels both NBs in the second instar larva. (G) Immortalization of R719B03[AD]-R45D04[DBD] marks lineage 20 and lineage 22 as postembryonic progeny of these NBs. Arrow and arrowheads indicate two separate bundles coming from two different lineages. (H) R24C10 marks medially located NB5-7 (arrows) but not NB5-4 (arrowhead). *gsb*-lacz (gray) and Msh (red) co-expression labels both NBs (G) Immortalization of R24C10 expression in NB5-7 progeny visualizes lineage 20, which extends a single bundle (arrows).

The following figure supplement is available for figure 6:

**Figure supplement 1.** (A-C) Late stage embryos shown.

identified only one NB, NB5-4, with such molecular expression (*Birkholz et al., 2013*) and proposed that lineage 20 and lineage 22 both arise from NB5-4 (*Birkholz et al., 2015*). Indeed, we identified one of these NBs as NB5-4 based on embryonic clones generated with the R19B05[AD]-R45D04[DBD] line. These clones contained a single NB and several motor neurons exiting the CNS via the SN as previously shown for the progeny of NB5-4 (*Figure 6A—figure supplement 1B*; *Schmidt et al., 1997*). However, clones from the other NB always appeared as a single large cell without any associated progeny. This pattern of one NB with progeny and the other without was evident in clones generated with R19B05[AD]-R45D04[DBD] and the GAL4 driver R17A10 (*Figure 6B—figure supplement 1C*). The lineage clones mentioned above tend to contain the entire progeny of the NB because we induced clone formation in gastrulating embryos. Thus, the isolated NB does not appear to make any embryonic progeny. Also, we never found the two NBs in the same clone, as would be expected if they came from the same precursor. Finally, we found the isolated NB only in the thoracic segments (*Figure 5D*). We believe this NB has not been previously described, so we named it NB5-7. The GAL4 line R24C02-GAL4 marks NB5-7 but not NB5-4 in the embryonic CNS, and its immortalization results in lineage 20 in the postembryonic CNS (*Figure 6I*).

We proposed that NB5-7 evolved from a duplication of NB5-4 in the thoracic segments for the following reasons: (i) NB5-7 and NB5-4 are similar in terms of their molecular marker expression; (ii) their postembryonic progeny (lineage 20 and 22) project their axons in a similar manner (*Truman et al., 2004*); and (iii) these two lineages were the most tightly correlated among all postembryonic lineages with regard to enhancer expression, suggesting that they share the most genes in common (*Li et al., 2015*). In the adult, the neuronal activity of lineage 20 and lineage 22 regulates leg posture (*Harris et al., 2015*).Thus, it is likely that during the evolution of *Drosophila*, the increase in the number of these neurons via NB5-4 duplication provided finer control over locomotion.

## Segmental differences during neurogenesis

### Hox genes restricts NB2-3, NB3-4 and NB5-7 to thoracic segments

Overall, we identified 30 paired NBs and one unpaired NB in the thoracic embryonic array. The number was reduced to 27 pairs as we moved into the abdomen with the loss of Msh expressing NBs-NB2-3, NB3-4, and NB5-7 (*Figure 5D*). It was possible that they formed in the abdominal segments but had very short lineages and were eliminated quickly by apoptosis. To investigate this possibility, we used H99 mutant embryos, which do not exhibit apoptosis (*White et al., 1994*). We did not detect any ectopic Msh$^+$ NBs in the abdominal segments of H99 mutant embryos; thus, it seems that these cells simply do not form in the abdominal segments (*Figure 7—figure supplement 1A,B*).

We next asked what factors are responsible restricting these NBs to the thoracic segments. HOX proteins specify segment identity and control segment-specific structure formation. We used R77B09 line, which marks NB2-3 and NB3-4, to examine whether HOX genes control formation of these NBs. Antennapedia (Antp) is necessary for the proper development of thoracic segments (*Wakimoto and Kaufman, 1981*). In *antp* mutant embryos, NB2-3 and NB3-4 lineages formed normally, but their axons failed to exit the CNS, especially in the T2 and T3 segments (*Figure 7B,C*). In agreement with this phenotype, in wild-type animals, we detected significantly higher level of Antp protein in the T2-T3 segments compared to the T1 segment (*Figure 7A*). Although there is an overall reduction of Antp in the T1 segment, Antp levels in these lineages are higher than in surrounding lineages (*Figure 7A*). During postembryonic development, Antp function is similarly necessary to direct leg motor neurons to innervate their target muscles (*Baek et al., 2013*).

Ultrabithorax (Ubx), another Hox protein, is excluded from NB2-3 and NB3-4 lineages and most of the thoracic segments; however, its expression reaches maximum at the posterior edge of the T3 segments and most of the A1 segments (*Figure 7D*). In *ubx* mutant embryos, we detected ectopic NB2-3 and NB3-4 lineages in the A1 segments (*Figure 7F*). Previous studies have shown that Ubx functions together with Abdominal-A (Abd-A) to specify abdominal fates in the nerve cord and that in embryos mutant for both *ubx* and *abd-A,* almost all abdominal segments acquire thoracic fates (*Prokop and Technau, 1994*). In agreement with these findings, we detected ectopic NB2-3 and NB3-4 lineages in A1-A8 segments of embryos lacking both *ubx* and *abdA* (*Figure 7G*). Moreover, when we stained these double mutant embryos against Msh, we detected NB5-7 as well as NB2-3 and NB3-4 in the abdominal segments (not shown). Therefore, as expected, the Hox genes serve to restrict NB2-3, NB3-4 and NB5-7 to the thoracic segments.

Since NB2-3 and NB3-4 generate leg motor neurons, their presence only in the thoracic segment is not surprising. However, NB4-4, which also produces only leg motor neurons postembryonically (lineage 24; *Figure 4—figure supplement 1H*), is present in both abdominal and thoracic segments, but this stem cell also produces a variety of interneurons during its embryonic phase.

### Segment-specific postembryonic survival of NBs

The number of embryonic NBs that also have a postembryonic neurogenic phase differs between the segments. Previous studies suggested that larval T2 segment contains the greatest number of these NBs and that the number of segmental NBs decreases as one moves anteriorly or posteriorly by the loss of members from this T2 set (*Truman et al., 2004*, *Figure 1—figure supplement 1*). We find, however, that NB5-6 provides an exception to this rule.

NB5-6 is eliminated in the thoracic and abdominal segments late in embryogenesis (*Baumgardt et al., 2009*). To confirm this, we immortalized *lbe*-GAL4 expression, which uniquely marks NB5-6 (*Figure 8A–B*). As expected, we did not observe any postembryonic lineage arising from NB5-6 in the thoracic and abdominal segments, but detected a postembryonic lineage in the S3 segment of the subesophageal ganglion (SEG) (*Figure 8C*). Neurons in this postembryonic lineage expressed Eyes Absent (EyA) and extended axons in a similar manner to EyA$^+$ embryonic NB5-6 progeny, which are born in the "Cas" window during late embryonic stages (*Figure 8A–C*; *Baumgardt et al., 2009*). We concluded that NB5-6 survives in the S3 segment during post embryogenesis to generate a postembryonic lineage. This conclusion is further supported by forcing NB5-6 to survive in the thorax by expressing the apoptosis suppressor p35 in this lineage. This resulted in an ectopic postembryonic lineage in the thoracic set and this lineage shared the position and projection pattern seen in its S3 counterpart.

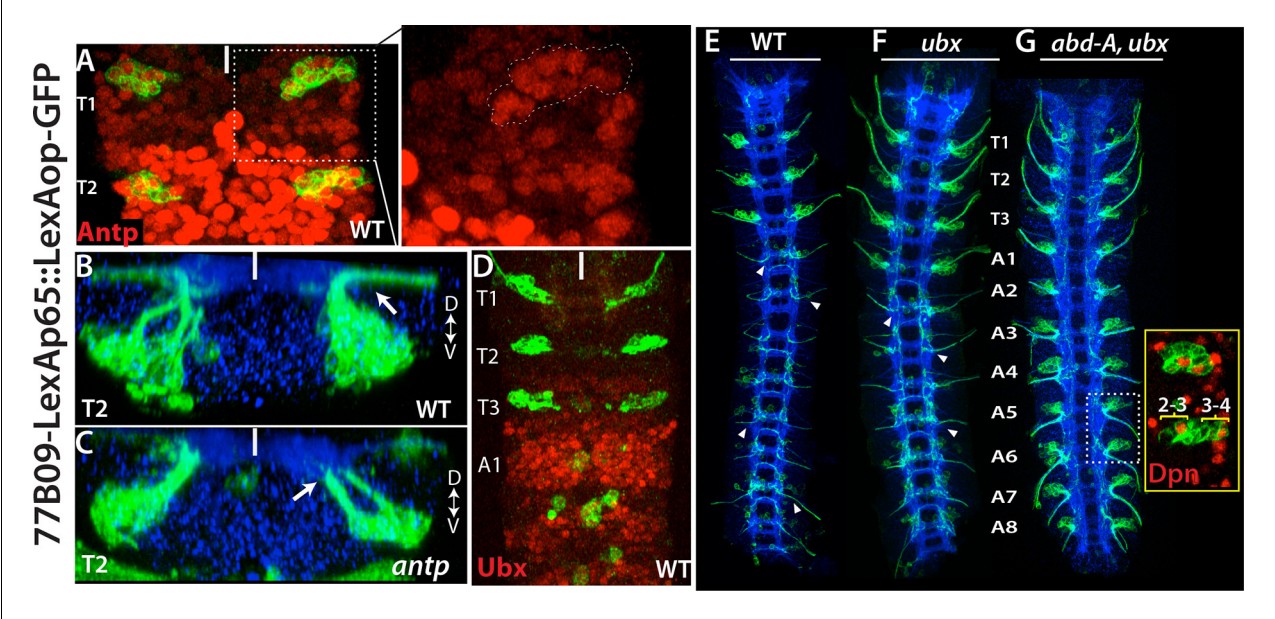

**Figure 7.** Hox genes restrict NB2-3 and NB3-4 lineages to the thoracic segments. (A-G) 77B09-lexAp65 driven GFP marks NB2-3 and NB3-4 lineages in wild type embryos (A, B, D and E) and embryos mutant for *antp* (C), *ubx* (F), and double mutant for *abd-A* and *ubx* (G). All embryos are at stage 16. (A) Antp expression (red) is significantly higher in T2-T3 segments than T1 segments (only T1 and T2 segments shown). (Inset) in the T1 segment, NB2-3 and NB3-4 progeny (outlined by dashed lines) express Antp at higher levels compared to other lineages. (B-C) Transverse view of the T2 segment showing that axons of NB2-3 and NB3-4 progeny exit the CNS in a wild type embryo (B) but fail to do so in an *antp* mutant embryo (C). (D) Highest Ubx expression (red) in a wild type embryo is detected around the A1 segments and thoracic segments virtually lack Ubx expression. (F-G) whole nerve cords from embryos with indicated genotypes shown. (E) 77B09-lexAp65 marks thoracic specific NB2-3 and NB3-4 lineages and in the abdominal segments it marks neurons of NB2-1 and NB7-3. Note also axons of sensory neurons entering the nerve cord in all abdominal segments (some marked by arrowheads) and these axons are thinner compared to exiting motor axons in the thoracic segments. (F) In embryos that are mutant for *ubx*, ectopic NB2-3 and NB3-4 lineages form in the A1 segment. (G) In embryos that are double mutant for *ubx* and *abd-A*, ectopic NB2-3 and NB3-4 lineages form in all abdominal segments except the terminal segment. Inset shows a magnified view of hemisegments of A5-A6 where Dpn (red) marks ectopic NB2-3 and NB3-4. WT, wild type; White bar, midline.

The following figure supplement is available for figure 7:

**Figure supplement 1.** Apoptosis is not responsible for the lack of NB2-3, NB3-4 and NB 5–7 in the abdominal segments.

The postembryonic lineage of NB5-6 that we detected in S3 segments was previously assigned to lineage 5 (NB5-3) based on morphological similarities to thoracic lineages (*Kuert et al., 2014*); however, the above results and our previous work disagree with this designation (*Li et al., 2014*).

## Neurons produced during embryonic versus postembryonic neurogenesis

The NBs of insects with incomplete metamorphosis, like grasshoppers, generate all of their progeny during embryogenesis (*Bate, 1976*; *Shepherd and Laurent, 1992*). With the evolution of metamorphosis, this single phase of neurogenesis was split into two, with the embryonic phase producing the neurons of the larva and the postembryonic phase dedicated to making adult neurons. We wanted to know if NB arrest late in embryogenesis simply interrupts the temporal progression of neural types or if the arrest somehow "reprograms" the NB so that cell classes produced after the arrest bear no resemblance to those produced before.

### NB2-3 produces leg motor neurons during both its embryonic and postembryonic neurogenic phases

The clearest example of a continuity of generating a cell class through the arrest is seen for NB2-3, which produces the postembryonic cluster of leg motor neurons constituting lineage 15. In the larva,

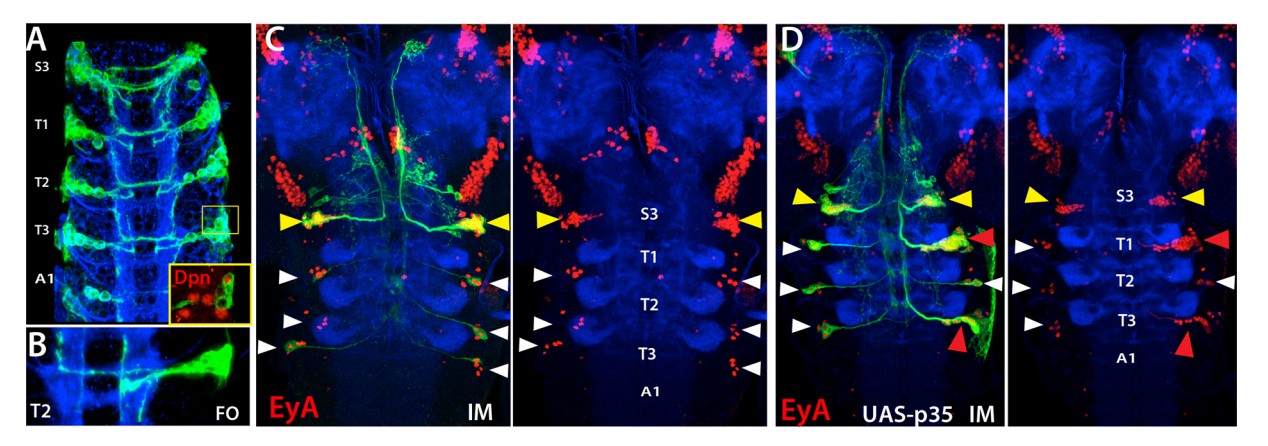

**Figure 8.** NB5-6 generates postembryonic progeny only in the S3 segments. Expression of lbe-K-Gal4 in the embryonic nerve cord (A-B) and immortalization of its expression in NB5-6 progeny with (C) or without p35 in the larval CNS (B). (A) S3-A1 segments of stage 16 embryos showing *lbe-K*-Gal4 driven GFP expression, which specifically marks Dpn+ NB5-6 (red in inset) and its progeny. (B) NB5-6 lineage clone from a stage 17 embryo extend axons in several different directions. (C) Immortalization of lbe-K-Gal4 expression marks NB5-6 postembryonic lineage (yellow arrowhead) in S3 segments and late-born embryonic neurons (white arrowhead) of NB5-6 in thoracic segments; NB5-6 does not generate a postembryonic lineage in thoracic segments. (D) When apoptosis is blocked by p35 misexpression, NB5-6 generates ectopic postembryonic progeny (red arrowheads). Both endogenous S3 and ectopic thoracic postembryonic progeny of NB5-6 express EyA (red) and extend axons in a similar manner to EyA+ neurons that are born in the embryonic "Cas" window of NB5-6 (Baumgardt et al., 2009; white arrowheads). HRP (blue) in embryos (A, B) and Phalloidin (Red) in larvae (C, D) visualize neuronal architecture. FO, flip-out lineage clone; IM, Immortalization.

these neurons arrest after extending their axons into the periphery towards the leg discs and wait until metamorphosis before invading the developing leg disc (*Figure 5C*, *Truman et al., 2004*; *Baek and Mann, 2009*; *Brierley et al., 2009* and *2012*). Like their postembryonic counterparts, the embryonic progeny of NB2-3 are primarily arrested motor neurons that lack muscle targets at hatching. We identified two lines, R77B09AD-R28H10DBD and VT006878, which mark ventrally and dorsally located NB2-3 embryonic progeny, respectively (*Figure 9D*). These arrested motor neurons are evident in these lines at hatching. By feeding larvae on food containing EDU from hatching and then examining the clusters just prior to metamorphosis (*Figure 9A,E*), we found that all of the arrested motor neurons lacked EDU incorporation, showing that these lines do not add any of the postembryonic members of this lineage. Their axons split into two bundles after exiting the larval CNS. One bundle extends across the body wall, and, in T1 only, includes one functional motor neuron that innervates a muscle field while the rest of the axons appear to stall in their peripheral nerve (not shown). The other bundle extends towards the leg disc but its axons also stall out on the nerve (*Figure 9C,F*). Except for the T1 motor neuron, these stalled neurons are also negative for vGLUT expression (a marker of mature motor neurons) (*Figure 9B*). Thus, our results suggest that almost all of the embryonic progeny of NB2-3 are developmentally arrested and a subset are potentially leg motor neurons.

We immortalized the expression of *VT006878* into the adult stage (see Materials and methods) to assess the fates of the arrested motor neurons. VT06878 marks 7.8 +- 0.6 (n=10) neurons per thoracic hemisegment in the larval VNC (*Figure 9G*). Immortalized expression marked 6.2+-0.7 (n=6) neurons per hemisegment in the adult VNC, and all of these neurons expressed vGLUT and extended axons into the leg (*Figure 9I* and not shown), where they innervated muscles of the coxa, trochanter and femur (*Figure 9J–L*). Consequently, it appears that for NB 2–3, its arrest occurs early in its program of making adult leg motor neurons resulting in a few being made prior to the arrest and the remainder after.

The embryonic progeny of NB3-4 also appear to be developmentally arrested motor neurons (*Figure 5J–M*), but we lacked the proper reagents to investigate their fate.

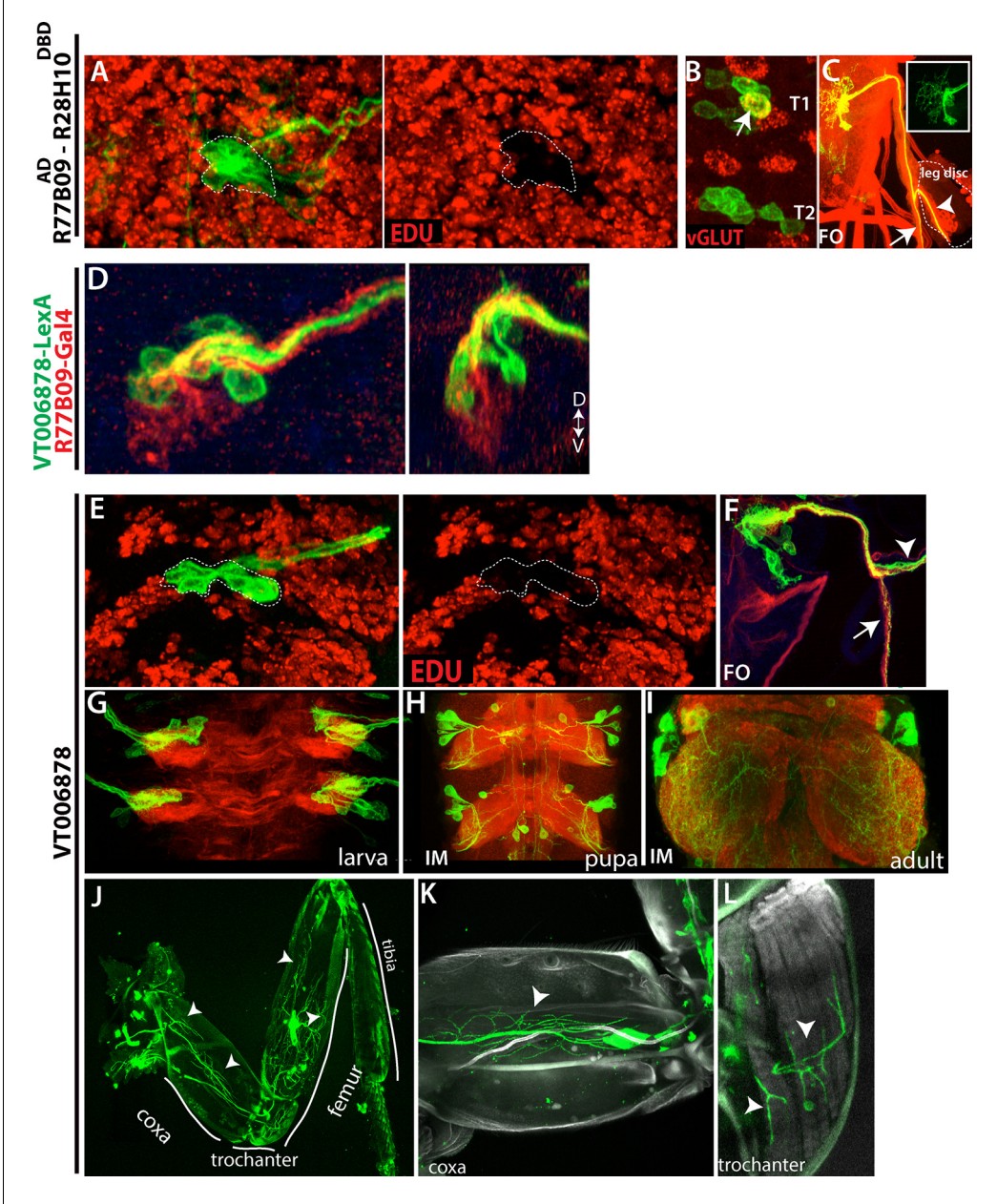

**Figure 9.** Embryonic progeny of NB2-3 are developmentally arrested during larval life, but differentiate during metamorphosis to be leg motor neurons. (A, E) R77B09AD-R28H10DBD (A) and VT006878 (E) expression in T2 hemisegments from a late stage larva shown. EDU (red) labels all of the postembryonic neurons but not NB2-3 progeny (outlined with dashed lines); hence outlined cells are born embryonically. (B) None of these embryonic-born cells marked by R77B09AD-R28H10DBD express vGLUT (indicator of mature motor neurons) with the exception of a motor neuron in the T1 segment (arrow). VT006878 marked neurons do not express vGLUT either (not shown). (D) R77B09 (red) and VT006878-LexA (green) mark ventral and dorsal embryonic progeny of NB2-3, respectively. Transverse view shown on the right; dorsal up. (C, E) T2 lineage clones, which are generated via R77B09AD-R28H10DBD(C) and VT006878 (E) are induced in an early embryo and visualized in the late-stage larva. Like in the embryo (see *Figure 5*), axons of NB2-3 embryonic progeny split into two bundles after exiting the CNS. One of these bundles extends toward the leg disc (arrowheads). (E) VT006878 expression in T2-T3 segments in the larval CNS. (H-L) Immortalization of VT006878 expression visualizes these neurons in the pupal (H) and adult nerve cord (I) and innervation pattern of their axons in the adult leg (J-L). (J) VT006878 marked neurons innervate muscles in the coxa, trochanter, and femur parts of the leg but not the tibia. High magnification views from coxa (K) and trochanter (L).

## Larval-born neurons follow the axonal path of late-born embryonic interneurons

The motor neurons are unusual in that even the embryonic-born members of the class are maintained in an arrested state until the start of metamorphosis. A similar pattern is also seen for one of the flight motor neurons (*Consoulas et al., 2002*). The vast majority of embryonic interneurons, though, become functional neurons in the larva, but some of these neurons still share characteristics with their postembryonic-born siblings (e.g., *Figure 3*). For example, axons of a subset of NB6-1 embryonic progeny bundle with the lineage 12A neurons (the postembryonic progeny of NB6-1), and both populations of neurons express Unc-4 (*Figure 10A* and not shown). Interestingly, in complex embryonic lineages with several different axonal projections, the postembryonic born neurons followed only one or two of these projections. For example, the embryonic progeny of NB4-1 are quite complex, with four main types of neurons: (i) efferent neurons, (ii) intersegmental interneurons, (iii) local interneurons that cross the midline via the posterior commissure, and (iv) local interneurons that cross the midline via the anterior commissure (*Bossing et al., 1996*; *Schmid et al., 1999*). The last class can be further subdivided based on whether they cross in dorsal or ventral neuropil (*Figure 10B*). The postembryonic progeny of NB4-1 conform to the two subclasses of the type-iv interneurons: axons from the 14A hemilineage cross the midline via the ventral anterior commissure, whereas axons of the 14B hemilineage use the dorsal anterior commissure (*Figure 10D*). We suspected that the postembryonic cells shared features with the last class of embryonic neurons to be born. To identify when these embryonic neurons are born, we roughly mapped their birth order of the NB4-1 progeny based on clone induction time and clone size. The smallest embryonic clones consistently contained only the two type-iv subtypes (*Figure 10C*), showing that they are the last type of neurons that are produced in the embryo. Cells with these two projection patterns then continue to be made through the entire postembryonic phase to generate the 14A and 14B hemilineages. Most of the 14B neurons, though, die soon after their birth (*Truman et al., 2010*).

We observed the same pattern in the NB3-2 and NB5-3 lineages, in which neurons born in the last embryonic divisions showed similar axonal projections to those of the postembryonic-born siblings. For example, during embryogenesis, NB3-2 generates two distinct sets of motor neurons (BarH$^+$ and Hb9$^+$motor neurons) and contralaterally projecting interneurons (*Figure 10E* and not shown; *Landgraf et al., 1997*; *Garces et al., 2006*). The postembryonic progeny of NB3-2, lineage 7, have similar axonal projections and bundle with the embryonic-born interneurons (*Figure 10J,K* and not shown). Moreover, both embryonic- and larval-born interneurons express Unc-4, but motor neurons do not. We mapped the birth order of NB3-2 embryonic progeny and found that motor neurons (first BarH$^+$, then Hb9$^+$) are born from the early cell divisions, while Unc-4$^+$ interneurons are born from the late embryonic divisions of NB3-2. Thus, NB3-2 generates Unc-4$^+$ interneurons before entering quiescent state in the embryo and then resumes generating Unc-4$^+$ interneurons with similar morphology in the larval stages.

For the NB3-3 lineage, Castor expression marks a temporal neurogenesis window that lasts from late embryonic to mid-larval stages (*Tsuji et al., 2008*). Indeed, we found that Cas expression is maintained in many NBs including NB3-2 during these stages (not shown). We hypothesized that both embryonic and postembryonic Unc-4$^+$ interneurons are born in the Cas window and that these neurons would therefore not form in animals lacking *cas* function, since Cas is necessary for the formation of neurons born during the embryonic Cas window (*Kambadur et al., 1998*; *Isshiki et al., 2001*). As expected, embryonic-born Unc-4$^+$ interneurons do not form in *cas* mutant embryos (*Figure 10H–I*), indicating that these neurons indeed arise in the Cas temporal window and that Cas function is necessary for their formation. However, the larval-born Unc-4$^+$ interneurons in *cas*-null MARCM clones formed and extended their contralateral axon similar to those of wild type (not shown), indicating that larval born neurons do not require Cas function for their identity and that temporal identity in these two phases of neurogenesis may be regulated differently.

We found a similar scenario in the lineages of NB5-3 and NB5-6, where neurons born in the Cas$^+$ embryonic window extend axons whose route is then later followed by the postembryonic progeny of these NBs (*Figure 10—figure supplement 1* and *Figure 8*). Thus our findings suggest that NBs generate similar types of neurons before and after the quiescent state, although these become incorporated into circuitry to control very different types of bodies, larval versus adult.

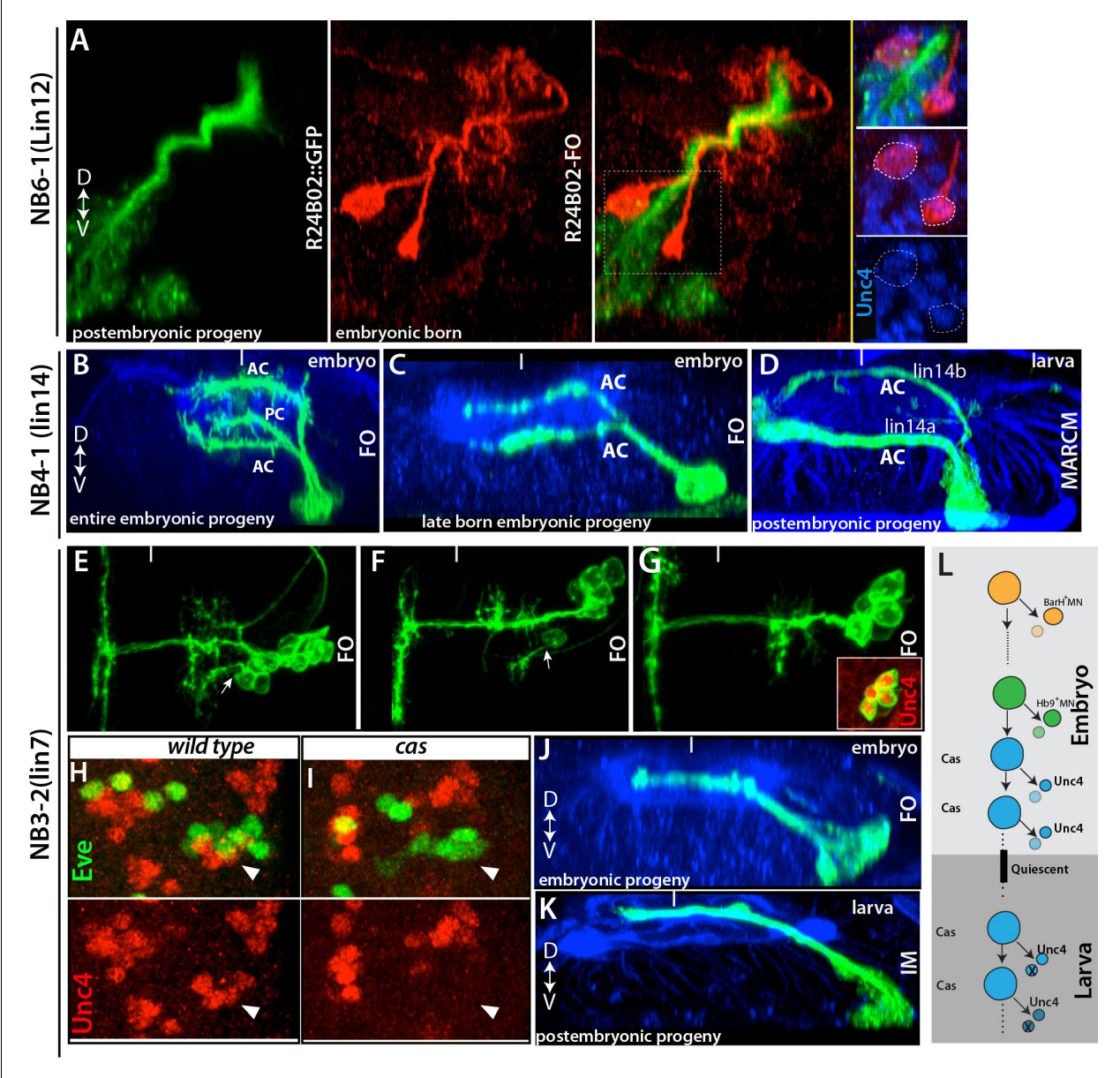

**Figure 10.** Late-born embryonic neurons and postembryonic neurons show significant similarities. (**A**) A transverse view of the larval thoracic nerve cord. R24B02 driven GFP is in green; flip-out clones, which are induced in the embryo via R24B02, are in red. Within the NB6-1 lineage, mature axons of the embryonic-born neurons (red cells) bundle with the immature axons of the larval-born neurons (lineage 12A neurons; green cells). Both embryonic and postembryonic-born neurons express Unc-4 (blue; inset). Note that the reporter driven via flip-out shows stronger signal than R24B02 driven GFP; thus, it visualizes embryonic progeny of NB6-1. (**B, C**) Transverse views from stage 17 embryos shown. R78A08 used to generate embryonic NB4-1 lineage clones. (**B**) An early-induced lineage clone showing the entire embryonic progeny of NB4-1. (**C**) Late induced lineage clone showing late-born embryonic progeny of NB4-1. (**D**) A MARCM clone from a larva showing that postembryonic progeny of NB4-1 extend their axons contralaterally in the ventral anterior commissure (lineage 14A) or dorsal anterior commissure (lineage 14B) in a similar manner to late born embryonic neurons in the lineage. (**E-G**) Nerve cords from stage 17 embryos shown. R21E09 used to generate embryonic NB3-2 lineage clones, which were induced at different time points (**E**, earliest; **G**, latest). (**E-F**) Earlier induced clones contain motor neurons (arrows). (**G**) The late induced clone contains only interneurons, which express Unc-4 (inset) and extend contralateral axons. (**H-I**) A single thoracic hemisegment from wild-type (**H**) and *cas* mutant (**I**) embryos shown. Both embryos are at stage17. (**H**) Unc-4 [+] neurons of NB3-2 (red) reside adjacent to Eve[+] NB3-3 neurons (green). (**I**) In a *cas* mutant embryo, Unc-4[+] neurons of NB3-2 are not detected. (**J-K**) Transverse views of embryonic and postembryonic progeny of NB3-2 shown. Late-born Unc-4[+] embryonic neurons in the embryonic CNS (**J**) and Unc-4[+] postembryonic neurons in the larval CNS (**K**) use the same route to cross the midline, anterior intermediate commissure. (**E-G, J**); immortalization of R21E09[AD]-R16H11[DBD] marks lineage 7 (**K**). (**L**) Schematic representation of NB3-2 neurogenesis from embryonic to larval stages. NB3-2 generates motor neurons and presumably sibling interneurons in early cell divisions. NB3-2 later produces Unc-4[+] contralateral interneurons in the Cas window. At the end of embryogenesis, it enters quiescent state. At the beginning of second larval stage, NB3-2

*Figure 10 continued on next page*

*Figure 10 continued*

resumes cell division and generates more Unc-4[+] contralateral interneurons in the Cas window. FO, lineage clone; IM, immortalization. Dorsal is up for images showing transverse views; for the rest anterior is up. HRP (blue) visualizes embryonic neuronal architecture (**A**, **C**, and **J**). BP104 (blue; **D**) and Phalloidin (blue; **K**) visualize larval neuronal architecture. FO, flip-out lineage clone; IM, Immortalization; White bars mark the midline.

The following figure supplement is available for figure 10:

**Figure supplement 1.** (**A-C**) NB5-3 lineage clones, which were induced at different time points (**A**, earliest; **C**, latest).

## Late-born embryonic neurons and postembryonic lineages show similar patterns of transcription factor expression

As noted above for a few lineages, embryonic and postembryonic progeny of a given NB express similar transcription factors and show similar axonal projections. Moreover, the late-born embryonic neurons are the most similar to the postembryonic group. We examined whether this pattern of common transcription factor expression is universal, by assaying all lineages for the expression of transcription factors at both embryonic and postembryonic stages (*Lacin et al., 2014b*, data in this study). We performed antibody staining against these transcription factors in embryos with FLP-out clones that marked specific embryonic lineages. As shown in *Table 2*, we found that, in almost all lineages, transcription factors that mark a specific postembryonic lineage are also expressed in some of the embryonic progeny of the same lineage. Moreover, neurons expressing these transcription factors in the embryo tended to be located nearest to the NB, indicating they are born late in the embryonic lineage. For example, the first division of NB4-2 generates Eve[+] RP2 and Dbx[+] RP2-sib neurons (*Lacin et al., 2009*). Later divisions generate Hb9[+] CoR motor neurons with their Dbx[+] sibling interneurons, followed by more interneurons (*Landgraf et al., 1997*; *Schmid et al., 1999*; *Lacin et al., 2009*). Based on our lineage clones, we found that these late-born neurons express D and Dbx (*Figure 11A–C*). Interestingly, the expression of D and Dbx in these late-born neurons is

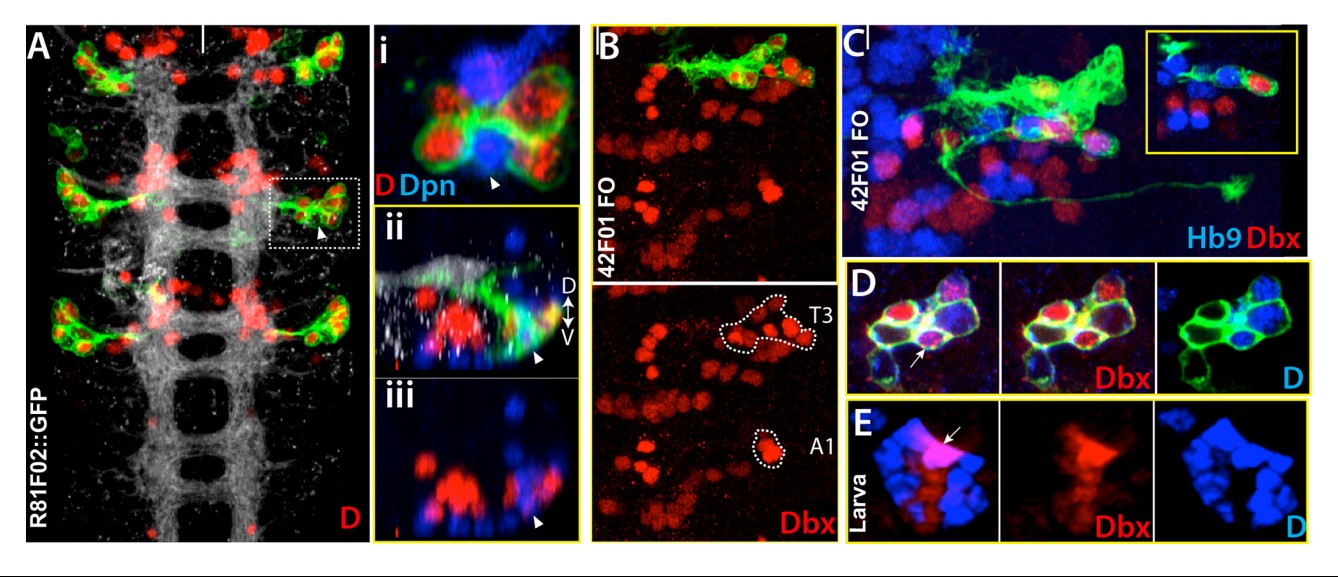

**Figure 11.** Dicheaete and Dbx marks both embryonic ands postembryonic NB4-2 progeny. Nerve cord images from the embryo (**A-D**) and larva (**E**). (**A**) R81F02 drives reporter expression only in thoracic segments. R81F02 marks D[+] neurons (red) of NB4-2; abdominal segments do not have these D[+] cells. (**i**) A single confocal section from the outlined region. D[+] cells reside adjacent to its stem cell, NB4-2 (arrowhead). (**ii**, **iii**) A transverse view of the same region. The NB and D[+] cells are on the ventral surface of the nerve cord. (**B-D**) NB4-2 lineage clones in the thoracic embryonic nerve cord. R42F01 used to generate the clones (**B**) Dashed lines outline Dbx[+] cells of NB4-2 lineage in T3 and A1 segments. Like D[+] cells, some of these Dbx[+] cells are present only in thoracic segments. (**C**) Hb9[+] motor neurons are adjacent to Dbx[+] neurons. (**D-E**) Dbx (red) and D (blue) are expressed in NB4-2 embryonic progeny (**D**) and postembryonic progeny, lineage 13 (**E**). FO, flip-out lineage clone. White bars mark the midline. Anterior is up. Dorsal is up in transverse view.

restricted to thoracic lineages and is carried over to the postembryonic progeny of NB4-2, with Dbx being expressed in hemilineage 13A and D expressed in hemilineage 13B (*Figure 11D–E*; *Lacin et al., 2014b*).

We also identified molecular markers with which we can uniquely identify most NBs from embryonic to larval stages (*Table 2*). Interestingly, some of these transcription factors are expressed exclusively in the NB of some lineages while in other lineages they are expressed in the progeny but not in the NB. For example, Dbx is expressed in NB3-2 during embryonic and postembryonic neurogenesis but not in its progeny. On the other hand, Dbx expression in the NB4-2 lineage is restricted to the progeny. Moreover, we also found that expression of NB patterning genes such as Gsb, Mirror, Ey, and Unpg is maintained from embryonic to larval development (*Table 2*).

## Conclusion

The segmental NB array in *Drosophila* embryos has been the focus of many studies to investigate neurogenesis. Individual NBs have been identified based on their large size and location in the array, by molecular marker expression including genetic handles such as lacZ and GAL4 lines, or by the morphology of their progeny. Only a few NBs have been studied comprehensively including all these features [e.g., NB5-6 (*Baumgardt et al., 2009*). Here, we have used a suite of molecular markers and a library of GAL4 lines, many of which are specific to individual NBs, to characterize the entire set of thoracic NBs, including their embryonic and postembryonic progeny.

We found that 26 of 30 (+1 medial) thoracic NBs survive into larval stages to generate neurons for the adult nervous system. We found that three of these NBs appear to be confined to the thorax, and two of them (NB2-3 and NB3-4) are dedicated to producing motor neurons for the leg. The third NB, NB5-7, had not been previously described and is unique in that it appears to have only a postembryonic neurogenic phase. Although the segmental NB array has been highly conserved through insect evolution (*Thomas et al., 1984*; *Truman and Bate, 1988*), we think that NB5-7 is a recent addition, likely by a duplication of NB5-4.

This two-phase pattern of neurogenesis evolved from a simpler scheme, such as seen in grasshoppers, in which all neurons were generated during an extended embryonic phase (Bate 1976; *Shepherd and Bate, 1990*). The similarity of the neurons just before and after quiescence in flies suggests that the insect NBs dealt with metamorphosis by simply suspending their ongoing fate determination through the arrest period and then resuming from that point once neurogenesis was restarted. In other words, metamorphosis did not likely require the resetting of a fate determination clock on the development of a novel postembryonic system.

## Materials and methods

### Fly Stocks

We used the following fly strains with indicated genotypes during this study Canton-S as wild type, *gsbn-lacZ.4z1* (*Li and Noll, 1994*), *lbe*(K)-*GAL4* (*Baumgardt et al., 2009*), *mirror-lacZ* (*Broadus et al., 1995*), *unpg-lacZ* (*Doe, CQ 1992*), *5172J-GAL4* (*Lacin et al., 2014a*), *elav-GAL4* (*DiAntonio et al., 2001*), *en- GAL4* (*Brand and Perrimon, 1993*), *ems-GAL4* (*Estacio-Gomez et al., 2013*), *eg*-kinesin-lacZ (*Higashijima et al., 1996*), *eg-GAL4* (*Ito et al., 1995*), *antp*[25] (*Abbott and Kaufman, 1986*), *Df(3L)H99* (*Abbott and Lengyel, 1991*), *Df(3R)Ubx109* (*Lewis, 1980*), *cas*[24]/*TM6b* (*Cui and Doe, 1992*), pJFRC19-13XLexAop2-IVS-myr::GFP, pJFRC7-20XUAS-IVS-mCD8::GFP (*Pfeiffer et al. 2010*).

### TUNEL and EDU labeling

Cell death was detected by TUNEL labeling (Roche, in situ cell death detection kit, TMR red). Embryos were first labeled with primary and secondary antibodies, and then treated with proteinase K (2 ug/ml) for 5 min at 37°C. TMR labeling was performed according to manufacturer's instructions.

To mark all cell born postembryonically, newly hatched first-instar larvae were fed yeast paste containing 300 μM EdU. At the wandering larval stage, larvae were dissected, fixed and stained as described previously (*Lacin et al., 2014b*).

## Lineage tracing

### Multi-Color Flipout generation

Embryos were collected for three hours from the crosses of flies carrying GAL4 lines with flies carrying hsFlp2::PEST;; HA_V5_FLAG (MCFO-1; *Nern et al., 2015*).

Embryos were incubated at 25°C for three or five hours and then dechorionated with bleach. To induce heat-shock mediated clone formation, embryos were floated in 60% glycerol on a 24X60-mm coverslip and incubated in a 37°C water bath for three to six minutes depending on the Gal4 expression. We then washed the embryos with wash buffer (*Patel, 1994*) and incubated them at 18 or 25°C in a Petri dish humidified with wet Kimwipes until the desired stage of fixation.

### Irreversible reporter expression

To drive reporter expression permanently in the progeny of a NB, flies carrying NB specific *GAL4* transgenes were crossed to flies carrying following transgenes: dpn>KDRT-stop-KDRT>Cre: PEST, lexAop-myrGFP;UAS-KD (*Awasaki et al., 2014*) and Act5c>loxP-stop-loxP>LexA::p65 (a gift from T Lee). Crosses were performed at 22°C for Gal4 lines and 25°C for split-Gal4 combinations.

We used *tub*-GAL80$^{ts}$ (*McGuire et al., 2003*) for *eg-GAL4*-mediated reporter immortalization to restrict this event only Eg-Gal4$^{+}$ embryonic NBs since it marked several other NBs after embryogenesis. To do that, we collected embryos for two hours at 25°C, incubated these embryos at 29°C until they reach first instar stage and then let them develop into late third instar stage at 18°C. Dissection and staining were performed before they become white pupae.

To irreversibly mark VT0048571-expressing neurons, we crossed flies carrying this *GAL4* driver with flies carrying *5XUAS-FlpPEST* (*Nern et al., 2011*). Progeny carrying both transgenes from this cross were mated with flies of the following genotype: w;13XLexAop2-IVS-myr::GFP, Actin5Cp4.6>dsFRT>nlsLexAp65;tub-gal80TS (*Pfeiffer et al. 2010*; *Harris et al., 2015*; *McGuire et al., 2003*). To immortalize only third-instar larval expression, progeny from this cross were kept at 18°C except between 100–112 hr after egg laying, when they were incubated at 29°C.

### MARCM clone generation

The MARCM technique was used to generate lineage clones (*Lee and Luo, 1999*). Newly hatched first instar larvae were incubated at 37°C for 30 min and nerve cords of late third instar larvae were dissected, fixed and stained. The following lines were used: *elav-Gal4; FRT42B, tub-Gal80* and *hs-FLP; FRT42B, UAS-mCD8-GFP* or *hs-FLP; FRT42B, UAS-mCD8-GFP* and *UAS-mCD8-GFP, FRT40A, act-Gal4.*

## Immunochemistry

Tissue fixation and staining were performed as described by Patel (1994). To visualize major axonal tracts we used AMCA-conjugated HRP or combination of BP102 and FasII in embryos, Alexa Fluor 568-conjugated Phalloidin or BP104 in larvae, and NC82 in adult flies. Secondary antibodies were obtained from Jackson Immunoresearch and Life Technologies and used at 1/200 dilution. Tissues were mounted in Vectashield (Vector labs). To get better resolution, some samples were cleared with xylene and mounted in DPX (*Truman et al., 2010*).

The following antibodies were used at indicated dilutions: Rabbit Runt (1/1000; E. Wieschaus), rabbit Vg (1/50; *Williams et al., 1991*), guinea pig Toy (1/500; U. *Walldorf*), guinea pig Ems (1/300; U. *Walldorf*), rabbit Msh (1/1000; *Isshiki et al., 1997*), rabbit Ey (1/1000; U. *Walldorf*), guinea pig Dbx (1/1000; *Lacin et al., 2009*), rabbit Hb9 (1/1000; *Broihier and Skeath, 2002*), guinea pig Hb9 (1/1000; *Broihier and Skeath, 2002*), rat Islet (1/500; *Broihier and Skeath, 2002*), guinea pig Lim3 (1/100; *Broihier and Skeath, 2002*), rat Nkx6 (1/500; *Broihier et al., 2004*), rabbit Unc-4 (1/1000; *Lacin et al., 2014b*), guinea pig Dpn (1/1000; J. Skeath), rabbit Dichaete (1/1000; *Nambu and Nambu, 1996*), rabbit Nmr1 (1/1000; *Leal et al., 2009*), rabbit anti-Eagle (1/500; *Dittrich et al., 1997*), rabbit Ubx (1/500; *Marin et al., 2012*) chicken GFP (1/500; # A10262, Life Tech.), Alexa Fluor 568 Phalloidin (1/250; # A12380, Life Tech.), rabbit HA (1/500; # 3724S, Cell Sig.), rat HA (1/500; 3f10, Roche), rat Flag (1/200; # NBP1-06712, Novus B.), Chicken V5 (1/500; # ab9113, Abcam), mouse B-gal (1/1000), rabbit B-gal (1/1000; # A11132, Life Tech.), Goat AMCA-HRP (1/200; # 123-155-021, Jackson L.). The following mouse primary antibodies were obtained from Developmental

Studies Hybridoma Bank: Engrailed-4D9 (1/5), BP104 (1/40), BP102 (1/100), FasII-D4 (1/100), NC82 (1/100), Antp-8C11 (1/20).

## Image analysis

A Zeiss LSM 710 was used to collect confocal images. Z projections were performed via FIJI (http://fiji.sc/Fiji): To the show presence of NB of interest in the clone, we projected Dpn channel by including only the sections that displayed this NB. Similarly, for channels in which neuronal architecture was shown by markers such HRP or BP102, we projected only the sections where the major bundles were located. In *Figure 10A*, a contaminating from a different segment was removed manually.

## Acknowledgements

We thank Y Cai, F Diaz-Benjumea, C Doe, M Noll, G Mardon, T Shirangi, and S Thor for fly strains. We thank S Carroll, C Doe, J Skeath, U Walldorf, and E Wieschaus for antibodies. We are grateful to G Rubin for split-GAL4 strains and B Dickson, T Lee, and B Pfeiffer for sharing reagents before publication. We thank D Miller, C Robinett, M Texada, I Siwanowicz and J Etheredge for helpful comments on this manuscript. We are indebted to T Laverty, K Hibbard, A Cavallaro and the Janelia Fly Core for fly husbandry, and A Howard for administrative support. This research is supported by HHMI.

## Additional information

### Funding

| Funder | Author |
| --- | --- |
| Howard Hughes Medical Institute | James W Truman |

The funders had no role in study design, data collection and interpretation, or the decision to submit the work for publication.

### Author contributions

HL, Conception and design, Acquisition of data, Analysis and interpretation of data, Drafting or revising the article; JWT, Conception and design, Analysis and interpretation of data, Drafting or revising the article

### Author ORCIDs

Haluk Lacin, http://orcid.org/0000-0003-2468-9618

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
