## [Decision Letter]

Thank you for submitting your work entitled "Lineage mapping identifies molecular and architectural similarities between the larval and adult *Drosophila* CNS" for consideration by *eLife*. Your article has been reviewed by three peer reviewers, and the evaluation has been overseen by K VijayRaghavan as the Senior Editor and the Reviewing Editor. All three reviewers, Chris Q Doe, Matthias Landgraf, and Heinrich Reichert, have agreed to share their names.

The reviewers have discussed the reviews with one another and the Reviewing Editor has drafted this decision to help you prepare a revised submission.

Summary:

This manuscript will be a landmark in the *Drosophila* neurogenesis field. It links *Drosophila* embryonic neuroblasts (eNB; which have a long history of descriptive and mechanistic studies) with larval neuroblasts (lNB; that also have a long history of descriptive and mechanistic studies). In this paper Haluk Lacin and James Truman present a range of new reagents, primarily Gal4 and split-Gal4 expression lines, that have allowed them to address key questions of *Drosophila* nervous system development. The authors identify three new neuroblasts that were previously missed. The manuscript corrects several eNB – lNB assignments made last year by Birkholz et al. (which was also very nice work). It shows that the last born eNB progeny are molecularly and morphologically similar to the lNB progeny, and thus the period of quiescence does not dramatically 'reprogram' the progenitor lineage. It describes the role of Hox genes and apoptosis in shaping the segment-specific lNB array (not super novel, but nice data).

In addition to having generated and mapped with molecular markers driver lines that will allow the wider community to work on specific neuroblast lineages in ways previously not possible, Lacin and Truman focus on several very interesting questions.

For example, working with expression lines, as opposed to manual or stochastic labelling approaches, they were able to determine the completeness of the neuroblast maps as published to date and discovered as yet uncharted neuroblasts: NB 5-7, which has the hallmarks of having potentially arise by duplication of NB 5-4 in thoracic neuromeres. Both the NB 5-4 and NB 5-7 postembryonic lineages (Lin22 and Lin20, respectively) are associated with leg motor control, suggesting this to be potentially a relatively late evolutionary modification of the groundplan of the neuroblast map.

In another example, they asked how the evolution of complete metamorphosis might have impacted on neuroblast division patterns. Here, the evidence presented suggest that with metamorphosis a temporal break has been inserted into the cell division pattern: late embryonic neurons are often similar in marker gene expression pattern and neurite trajectory to the early postembryonic cells. This suggests that postembryonically lineages resume where they left off, as opposed to starting generally with a novel programme of differentiation.

Essential revisions:

The main issue that needs to be addressed with the manuscript is that it still leaves the reader somewhat uncertain as to precisely where the discrepancies lie between this work and the study previously studied by the Technau lab. The authors present detailed data for 3 of these, but very little for the remaining 5. The authors certainly must have detailed data on these 5 and they should present it in the manuscript, especially since further experiments are not required for this. It will be important for the field that both in the text and in Table 2 discrepancies in interpretation are clearly outlined. For example, for lineage 15 it would be helpful to read explicitly that this study identifies its embryonic origin as NB 2-3, while Birkholz and colleagues had suggested lineage 15 as being generated by NB 3-2. While the authors give a clear account of why they think NB 2-3 as the correct assignment, it would help to also read as to why they think lineage 7 the postembryonic product of NB 3-2. In other words, it could be most helpful to the field if the authors could find a way of constructively reconciling discrepancies where possible, and to point to those where reasonable doubts remain. Before final acceptance it is essential that the authors discuss each of the 8 neuroblast lineage assignments, in which the authors differ from Birkholz et al., in detail. If they wish the authors could reduce the description of neuroblast lineages in which both studies concur.

The only 'negative' to this nice paper is the fact that a similar paper was published last year. However, this paper confirms most of their data using an independent method, corrects a few errors, and provides a wealth of new reagents that will be extremely valuable to the field. Thus, we feel the paper is a major advance. In summary, this is an impressive study that fills an important gap in our understanding, and is executed and presented with care.

---

## [Author Response]

The main issue that needs to be addressed with the manuscript is that it still leaves the reader somewhat uncertain as to precisely where the discrepancies lie between this work and the study previously studied by the Technau lab. The authors present detailed data for 3 of these, but very little for the remaining 5. The authors certainly must have detailed data on these 5 and they should present it in the manuscript, especially since further experiments are not required for this. It will be important for the field that both in the text and in Table 2 discrepancies in interpretation are clearly outlined. For example, for lineage 15 it would be helpful to read explicitly that this study identifies its embryonic origin as NB 2-3, while Birkholz and colleagues had suggested lineage 15 as being generated by NB 3-2. While the authors give a clear account of why they think NB 2-3 as the correct assignment, it would help to also read as to why they think lineage 7 the postembryonic product of NB 3-2. In other words, it could be most helpful to the field if the authors could find a way of constructively reconciling discrepancies where possible, and to point to those where reasonable doubts remain. Before final acceptance it is essential that the authors discuss each of the 8 neuroblast lineage assignments, in which the authors differ from Birkholz et al.

, in detail. If they wish the authors could reduce the description of neuroblast lineages in which both studies concur. The only 'negative' to this nice paper is the fact that a similar paper was published last year. However, this paper confirms most of their data using an independent method, corrects a few errors, and provides a wealth of new reagents that will be extremely valuable to the field. Thus, we feel the paper is a major advance. In summary, this is an impressive study that fills an important gap in our understanding, and is executed and presented with care.

We appreciate the favorable evaluation of this study and suggested revisions. To address the main concern of the reviewers, we included sections of detailed explanations for each lineage, whose NB assignment differs from Birkholz et al. (2015). Further, we added Table 3 to the manuscript to summarize these explanations for an easier identification and comparison of these differences for the readers.

The following paragraph was added to the section”NB2-4, NB2-5, NB3-5 and NB6-2 generate postembryonic lineages 18, 17, 9 and 19, respectively in the dorsal part of the nerve cord.”

“Our conclusion that NB2-4 is the parent NB for lineage 18 differs from Birkholz et al. (2015), which concluded NB3-4 generates lineage 18. The embryonic lineage clone that was used in their study to link NB3-4 to lineage 18 appears to be a partial NB2-4 clone, which lacks early-born motor neurons. Birkholz et al. (2015) identified lineage 8 as the postembryonic progeny of NB2-4. However, our findings based on lineage tracings and molecular markers show that NB3-3 is responsible for the lineage 8 neurons (see below).”

The following three sections were added to the manuscript.

1) “Postembryonic lineages 7 and 13 are produced by NB3-2 and NB4-2, respectively”

“NB3-2 and NB4-2 generate progeny with similar morphologies: both generate ipsilateral motor neurons and contralateral interneurons. […] Moreover, immortalization of another driver R21E09^AD^-R28H10^DBD^, which marks NB3-2 in the embryo, identifies lineage 7 as its progeny (Table 1).”

2) “Postembryonic lineage 8 is produced by NB3-3”

*“ems*-GAL4 marks NB2-2, NB3-3, and NB3-5 and its immortalization labels lineages 10, 8, and 9 (Figure 4—figure supplement 1; Estacio-Gomez et al., 2013; Moris -Sanz et al., 2014). […] We think their designation is unlikely since NB2-4 is located in the dorsal surface of the nerve cord (unlike NB3-3 and lineage 8), does not express Ems and generates the dorsally located lineage 18 (Figure 3).”

New panels were added to Figure 4—figure supplement 1 during the revisions to show that lineage clones we identified are indeed NB3-3 clones and Ems is expressed in lineage 8 NB. Moreover, we corrected a mistake in this figure: A lineage clone from eg-GAL4 was mislabeled as ems-GAL4. In the revised version, we included NB3-3 lineage clones from both eg-GAL4 and ems-GAL4. Nonetheless, correcting this mistake did not change the content of this figure.

3) “Postembryonic lineage 11 is produced by NB7-2”

“Reporter immortalization of 35B12^AD^-R28H10^DBD^ marks specifically lineage 11 (12 of 30 hemisegments; Figure 4). […] Moreover, *unpg*-Lacz, which labels NB7-2 but not NB6-4 (Doe, 1992) marks the NB of lineage 11 (Figure 4—figure supplement 3).”

Figure 4—figure supplement 3 was added during the revisions to illustrate unpg-LacZ expression in lineage 11 NB.